# Antiviral responses in a Jamaican fruit bat intestinal organoid model of SARS-CoV-2 infection

Marziah Hashimi[1], T. Andrew Sebrell[1], Jodi F. Hedges[1], Deann Snyder [1], Katrina N. Lyon[1], Stephanie D. Byrum [2,3], Samuel G. Mackintosh[2], Dan Crowley[1,8], Michelle D. Cherne[1], David Skwarchuk[1], Amanda Robison[1], Barkan Sidar[4,5], Anja Kunze [6], Emma K. Loveday[4,5], Matthew P. Taylor[1], Connie B. Chang [4,5,9], James N. Wilking[4,5,9], Seth T. Walk[1], Tony Schountz [7], Mark A. Jutila[1] & Diane Bimczok [1,5] ✉

Bats are natural reservoirs for several zoonotic viruses, potentially due to an enhanced capacity to control viral infection. However, the mechanisms of antiviral responses in bats are poorly defined. Here we established a Jamaican fruit bat (JFB, *Artibeus jamaicensis*) intestinal organoid model of severe acute respiratory syndrome coronavirus-2 (SARS-CoV-2) infection. Upon infection with SARS-CoV-2, increased viral RNA and subgenomic RNA was detected, but no infectious virus was released, indicating that JFB organoids support only limited viral replication but not viral reproduction. SARS-CoV-2 replication was associated with significantly increased gene expression of type I interferons and inflammatory cytokines. Interestingly, SARS-CoV-2 also caused enhanced formation and growth of JFB organoids. Proteomics revealed an increase in inflammatory signaling, cell turnover, cell repair, and SARS-CoV-2 infection pathways. Collectively, our findings suggest that primary JFB intestinal epithelial cells mount successful antiviral interferon responses and that SARS-CoV-2 infection in JFB cells induces protective regenerative pathways.

Bats are considered important natural reservoirs for a variety of emerging zoonotic viruses that cause several illnesses in humans and other mammals[1], including severe acute respiratory syndrome coronavirus (SARS-CoV), Middle East respiratory syndrome coronavirus (MERS-CoV), Hendra virus, and Nipah virus[2–6]. The COVID-19 pandemic was caused by severe acute respiratory coronavirus-2 (SARS-CoV-2)[7], which also is thought to have its evolutionary origin in bats. This hypothesis is based on multiple studies that demonstrated a high level of genetic similarity between SARS-CoV-2 and several bat-borne coronaviruses such as RatG13 (96.1% identity[7]) and BANAL-52 (96.8% identity[8]), which have been detected in bat feces. Studies from a number of different bat species have shown that bat viruses, including coronaviruses, achieve long-term colonization of intestinal tissues without causing apparent disease[9–12]. In a study by Watanabe et al. on wild bats captured in the Philippines[10], enteric coronaviruses were detected in >50% of the animals, but clinical signs of disease were absent. Similarly, Subudhi et al.

[1]Montana State University, Department of Microbiology and Cell Biology, Bozeman, MT, USA. [2]University of Arkansas for Medical Sciences, Department of Biochemistry and Molecular Biology, Little Rock, AR, USA. [3]Arkansas Children's Research Institute, Little Rock, AR, USA. [4]Montana State University, Chemical and Biological Engineering Department, Bozeman, MT, USA. [5]Center for Biofilm Engineering, Bozeman, MT, USA. [6]Montana State University, Electrical and Computer Engineering Department, Bozeman, MT, USA. [7]Department of Microbiology, Immunology, and Pathology and Center of Vector-Borne Infectious Diseases, Colorado State University, Fort, Collins, CO, USA. [8]Present address: Department of Public & Ecosystem Health, Cornell University College of Veterinary Medicine, Ithaca, NY, USA. [9]Present address: Department of Physiology and Biomedical Engineering, Mayo Clinic, Rochester, MN, USA. ✉e-mail: diane.bimczok@montana.edu

found that 30% of North American little brown bats harbored coronaviruses in their intestines but did not display any signs of illness[11]. Becker et al. describe a similar level of coronavirus infection, 21%, in rectal swabs of vampire bats (*Desmodus rotundus*), with no significant impact on serum proteome composition[13]. Tong et al. analyzed rectal swabs and intestinal tissues from asymptomatic fruit bats in Peru and identified a novel influenza A virus, H18N11[12]. In contrast to bats, where gastrointestinal infections with eukaryotic viruses are frequent and are commonly asymptomatic[14], a similar colonization of the human gut with non-pathogenic eukaryotic viruses has not been reported, pointing to species-specific mechanisms[15].

Studying coronavirus infection in the GI tracts of bats is difficult, since few institutions maintain bat colonies for in vivo infection experiments, and cell lines from the GI tract of bats are not available, limiting in vitro analyses[16,17]. Organoid cultures have excellent potential as a model to study the mechanisms of viral infection in bat cells in vitro[18,19], since organoid lines can be derived from multiple species and tissues and can be maintained long-term. Organoids are permanent three-dimensional cultures that replicate the physiological and functional characteristics of their tissues of origin and that allow controlled studies of complex primary GI epithelial tissues in vitro[20]. Organoids from various human and murine tissues have been developed from tissue-derived stem cells and have been successfully used to investigate a wide range of disease processes, including viral infections[18,19,21,22]. Importantly, growth conditions for organoids appear similar across multiple species[23]. In a recent study, Chan et al. successfully generated 3-D airway organoids from tracheal epithelial monolayer cultures of cave nectar bats, *Eonycteris spelaea*[24]. Two previous reports describe the generation of intestinal organoid cultures from bat species[25,26]. Intestinal organoids developed from Chinese horseshoe bats, *Rhinolophus sinicus*, showed susceptibility to SARS-CoV-2, but lacked long-term active proliferation past 4-5 weeks[27]. Intestinal organoids derived from Leschenault's rousette, *Rousettus leschenaultii*, showed susceptibility to *Pteropine* orthoreovirus, but not SARS-CoV-2[25]. However, neither of these studies evaluated the cellular antiviral mechanisms of bat organoid tissues[25,27].

The hypothesis that altered IFN responses in bats compared to other species promote increased viral tolerance is a central paradigm in bat immunology[28–30]. In Australian black flying foxes (*P. alecto*) and lesser short-nosed fruit bats (*C. brachyotis*), a high level of constitutive IFN-α expression was detected[30], which has led to the concept that an "always on" IFN signaling system in bats can effectively suppress viral replication and prevent disease early on after infection[28,31]. Increased basal gene expression in bats also was described for several other genes involved in innate viral recognition and response, including IRF1, IRF3 and IRF7[32] and the ISG oligoadenylate synthase 1 (OAS1)[33]. Conversely, gene expression of type I IFNs in multiple tissues from Egyptian fruit bats (*R. aegyptiacus*) was low at baseline, but was inducible upon viral infection[34]. Other studies have reported dampened activation of stimulator of IFN genes (STING), a nucleic acid sensor involved in the regulation of IFN expression upon viral infection[35,36]. These reports highlight that the mechanisms of IFN expression, regulation, and signaling appear to be unique to individual bat species, pointing to a need for more detailed analyses.

Jamaican fruit bats (*Artibeus jamaicensis*, JFBs) are thought to be natural carriers of zoonotic pathogens such as rabies virus, West Nile virus and dengue virus[37–40]. JFBs are New World bats that are among the most common bats in the Americas and often live close to human settlements, so that spillover of zoonotic pathogens may occur. JFBs also are susceptible to experimental infection with Zika virus and MERS-CoV[4,41]. Based on the recently annotated genome[42,43], JFBs have one interferon (IFN)-β, four IFN-α, one INF-κ, one IFN-ε and four IFN-ω genes. Multiple interferon regulatory factors (IRFs) have also been identified. Therefore, JFBs are considered a relevant and tractable model system for studies of viral infection and antiviral immunity.

Here we established and characterized gut organoids from JFBs to study the susceptibility and immune response of the JFB intestinal epithelium to SARS-CoV-2 infection. Importantly, our organoid model was developed for a New World bat species, which have generally been underinvestigated[44,45]. We found that JFB intestinal epithelial cells supported modest viral replication that did not result in the release of infectious virions or cytopathic effects. Importantly, the organoids mounted a robust interferon response following exposure with infectious SARS-CoV-2. Proteomics and pathway analysis revealed that the JFB organoid proteome profiles matched profiles found in other SARS-CoV-2 infection studies in human nasal epithelium and multiple cell lines[46–50]. Moreover, SARS-CoV-2 infection activated innate inflammatory and cellular repair responses in the intestinal organoid model.

## Results

### Development and characterization of JFB gastrointestinal organoids

We established JFB gastrointestinal organoid cultures from fresh and cryopreserved stomach and from proximal and distal intestine (Fig. 1a and Supplementary Fig. 1a, b). The microscopic anatomy of the proximal and distal intestinal tissue used for organoid derivation was consistent with that of the small intestine, with prominent villi and few goblet cells (Fig. 1b and Supplementary Fig. 3b). Organoids formed within one day of crypt/gland isolation and were successfully maintained in a simple growth medium containing DMEM and 50% L-WRN-conditioned medium (Supplementary Fig. 1c). The murine noggin, R-spondin, and Wnt3a secreted by the L-WRN cells[51] show protein sequence similarities of 98%, 86%, and 99% with the orthologous JFB proteins, suggesting that these factors would be active in JFB cells (Supplementary Fig. 2).

Established JFB organoids mimicked the epithelial structure of JFB gastrointestinal tissue, with a simple columnar epithelium, a basal nucleus, and a defined luminal space (Fig. 1b, Supplementary Fig. 3a, b). Mucus-secreting goblet cells were more common in organoids derived from distal intestine than those from proximal intestine, similar to the cellular composition of the respective tissues of origins (Fig. 1b, Supplementary Fig. 3a, b). Morphometric analysis with OrganoSeg[52] showed that organoid size varied between different passages, but with no clear trends, and organoid shape also did not change significantly over six consecutive passages (Fig. 1c).

We next performed transcriptional analysis of the organoids to confirm tissue-specific gene expression patterns. The distal and proximal intestinal organoids expressed the intestine-specific genes *Vil1* and *Cdx2* as well as *Ace2*, while the gastric organoids showed increased expression of the chief cell marker pepsinogen C (*Pgc*) (Fig. 1d). For two representative lines of the distal intestine, expression of the intestinal markers *Muc2*, *Vil1* and *Cdx2* remained relatively stable over eight passages (Fig. 1e), with a significant increase of villin expression in p7 but no clear trends overall. Expression of *Pgc* and *Ace2* also remained stable (Fig. 1e). While *Pgc* is predominantly expressed in the stomach, expression in the small intestine has been described in humans[53]. Notably, copy numbers for *Ace2* were very low compared to the other targets.

We focused our further analyses on organoids from the intestine as a putative site for SARS-CoV-2 replication. To confirm the identity of the distal intestinal epithelium in our organoid model, we used immunofluorescence staining with cross-reactive antibodies and reagents. The majority of organoids had a typical apical-in conformation, with apical villin expression and strong phalloidin staining of the apical cell portions, indicative of the terminal actin web and microvilli formation (Fig. 2a). All cells also expressed intracellular epithelial cytokeratin. ACE2 expression was detected on the apical cell surface, with some weak basal staining. Transmission electron microscopy (Fig. 2b) confirmed the presence of characteristic microvilli on the apical surface of the epithelial cells, along with electron-dense apical

junctional complexes consistent with an enterocyte phenotype. We further characterized the JFB distal intestinal organoids by performing an unbiased proteome analysis using data-independent acquisition (DIA) mass spectrometry. Several key proteins characteristic of small intestinal epithelial cells in other mammals such as villin, E-cadherin, keratin 18 and 19, Na+/K+ ATPase, claudin 18, and a mucin (MUC5AC-like) were detected (Fig. 2c)[54]. Measurement of transepithelial electrical resistance (TEER) across organoid monolayers seeded on transwell inserts showed that the gastrointestinal organoids established a physiological epithelial barrier, with the stomach having the highest TEER compared to the intestinal organoids (Fig. 2d). Collectively, these analyses demonstrate that gastrointestinal organoids from JFBs replicate key features of the gut epithelium.

### Infection of intestinal organoids from JFBs with SARS-CoV-2 leads to replication of viral genomes

To determine whether the JFB intestine supports SARS-CoV-2 infection, organoids were dissociated and then inoculated with SARS-CoV-2 at MOIs of 0.1, 1 and 10. We selected distal intestinal organoids for these experiments, based on several previous publications that demonstrated SARS-CoV-2 replication in human ileal organoids[55–57]. Quantitative PCR analysis of viral genomes in JFB organoid cell lysates revealed a significant, concentration-dependent increase (>1 log, $P \leq 0.05$) in SARS-CoV-2 gene E RNA at 48 and 72 h post infection (hpi, Fig. 3a). The SARS-CoV-2 PCR in culture supernatants showed a similar increase at an MOI of 1 at 48 hpi (Fig. 3b). Importantly, significant expression of subgenomic (sg)RNA (>2 log-fold above baseline) for gene E indicating active viral replication in the organoids also was identified[58], albeit at low levels (Fig. 3c). However, plaque assays performed on the culture supernatants failed to detect the presence of infectious SARS-CoV-2 above baseline values derived from the inoculum, suggesting incomplete or ineffective viral replication or failure to secrete infectious progeny virus (Fig. 3d). Notably, SARS-CoV-2 incubation in medium for 48 h did not impact detection of viral copy numbers by PCR but did reduce the viral titer measured by plaque assay by >1 log-fold, suggesting a loss of infectivity over time

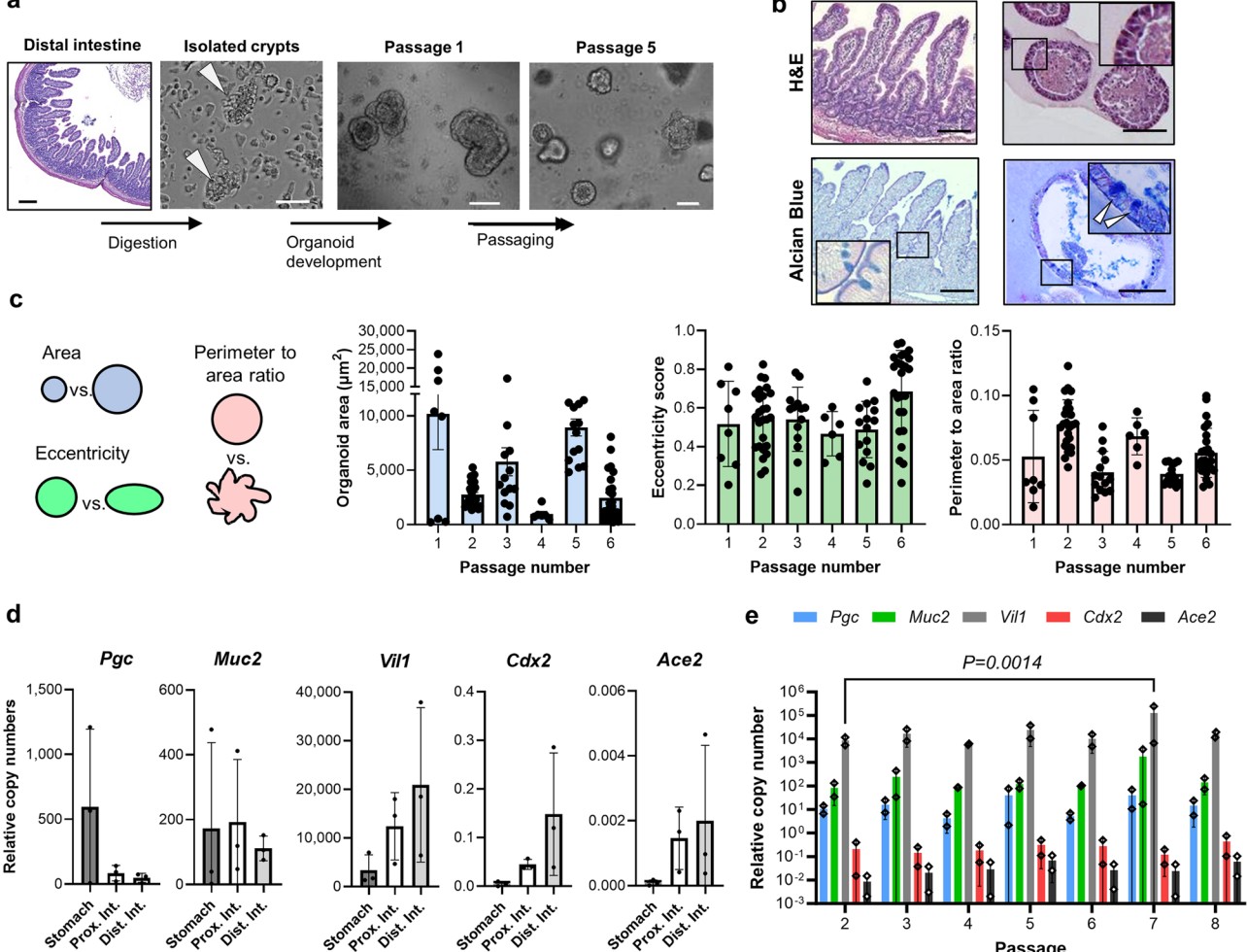

**Fig. 1 | Development and culture of gastrointestinal organoids from Jamaican fruit bats. a** Organoid derivation from Jamaican fruit bat (JFB) distal intestine. Tissue of origin, isolated intestinal crypts and formed organoids, representative of tissues from three bats, are shown. Scale bar: 200 µm for distal intestine, others are 50 µm. **b** Morphology of distal intestinal tissue (left) and distal intestinal organoids (right). Formalin-fixed, paraffin-embedded sections were stained with H&E (top row; bat 001, p2) or Alcian Blue (bottom row; bat 004, p3). High magnification insets show columnar cell shape and morphology of mucus-secreting goblet cells. Bars: 100 µm. Images are representative of 2 organoid lines and tissues. **c** Size and morphology of distal intestinal organoids were analyzed over six consecutive passages using OrganoSeg[52]. Dots: individual organoids (p1: $n = 8$, p2: $n = 26$, p3: $n = 14$, p4: $n = 6$, p5: $n = 15$, p6: $n = 25$); bars: mean ± SD. **d** Tissue-specific gene expression patterns in JFB organoids derived from stomach, proximal and distal intestine. Pooled qRT-PCR data from $n = 3$ established organoid lines (p2-5) are shown; mean ± SD. **e** Gene expression of distal intestinal organoids from two lines (bat004 and 005) was monitored over eight passages. Mean ± SD from two organoid lines with two technical replicates each; data was analyzed using ANOVA with Dunnett's multiple comparison test; $P = 0.0014$ compared to p2. Source data are provided as a Source Data file.

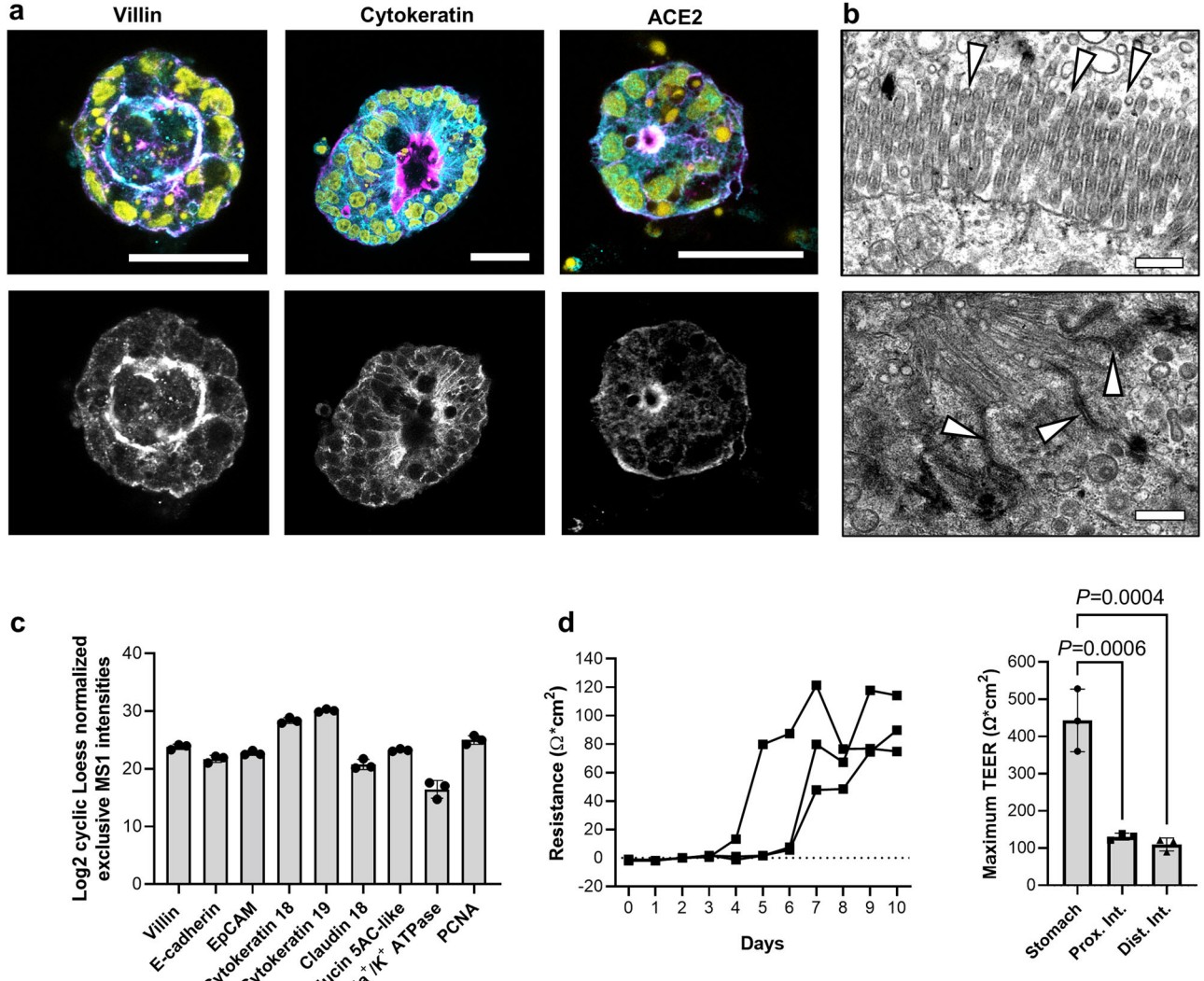

**Fig. 2 | Intestinal organoids from Jamaican fruit bats maintain key characteristics of the intestinal epithelium. a** Expression of intestinal epithelial cell markers and of ACE2 in JFB distal intestinal organoids. Whole mount cultures from three organoid lines (bat003, 004 and 005, all at p2) were stained with crossreactive antibodies to cytokeratin, villin and ACE2 (cyan), phalloidin-Ax488 (magenta) and DAPI (yellow) and were imaged by confocal microscopy. Representative Z stacks created from 3-5 adjacent images are shown. Bars: 25 μm. **b** Transmission electron microscopy images of JFB intestinal organoids, representative of two independent samples, show apical microvilli (top) and apical junctional complexes (bottom). Bat001, p1, bar = 500 nm **c** Expression of select intestinal epithelial cell-specific proteins. JFB distal intestinal organoids (bat003, p9, $n$ = 3 technical replicates) were lysed and processed for data-independent acquisition mass spectrometry. Individual datapoints and mean ± SD. **d** Transepithelial resistance (TEER) of JFB organoid cells cultured on transwell inserts for 10 days (bat001, p5). One representative experiment with triplicate wells of distal intestinal organoids (left), and comparative data for the highest TEER achieved by three independent cultures each of gastric, proximal, and distal intestinal organoids are shown (right; individual data, mean ± SD); one-way ANOVA with Tukey's post hoc test. Source data are provided as a Source Data file.

(Supplementary Fig. 4). Interestingly, immunofluorescence analysis of SARS-CoV-2 spike protein in infected JFB organoids revealed only a few positive cells, and these cells were not associated with morphologically intact organoids (Fig. 3e).

### Lack of cytopathic effect but Increased growth in SARS-CoV-2-infected JFB organoids

We also evaluated the cell viability of JFB distal intestinal organoids following SARS-CoV-2 infection by measuring caspase-3 activity with NucView® [59] (Fig. 4a). In Vero E6 cells, infection with SARS-CoV-2 induced a strong upregulation of caspase-3, consistent with the well-characterized cytopathic effect of the virus in this cell type. A small number of apoptotic cells were present in all JFB organoid cultures, likely due to physiological cell turnover. However, in contrast to observations in *Rhinolophus sinicus* organoids[27], SARS-CoV-2 did not

appear to have a cytopathic effect in JFB organoids (Fig. 4a, b), since the proportion of apoptotic cells did not change upon infection. Interestingly, SARS-CoV-2 caused a significant increase in organoid size and in the number of organoids that had re-formed from single cells after 48 h of infection (Fig. 4c, d), indicating that viral infection triggered increased cell proliferation in the bat intestinal epithelium.

### SARS-CoV-2 induces expression of type I interferons and proinflammatory cytokines in JFB organoids

Unique characteristics of the interferon (IFN) system have been linked to the increased viral tolerance observed in many bat species[28]. Therefore, we used quantitative RT-PCR to analyze gene expression of type I interferons and proinflammatory cytokines in JFB distal intestinal organoids following 48 h exposure to SARS-CoV-2. As shown in

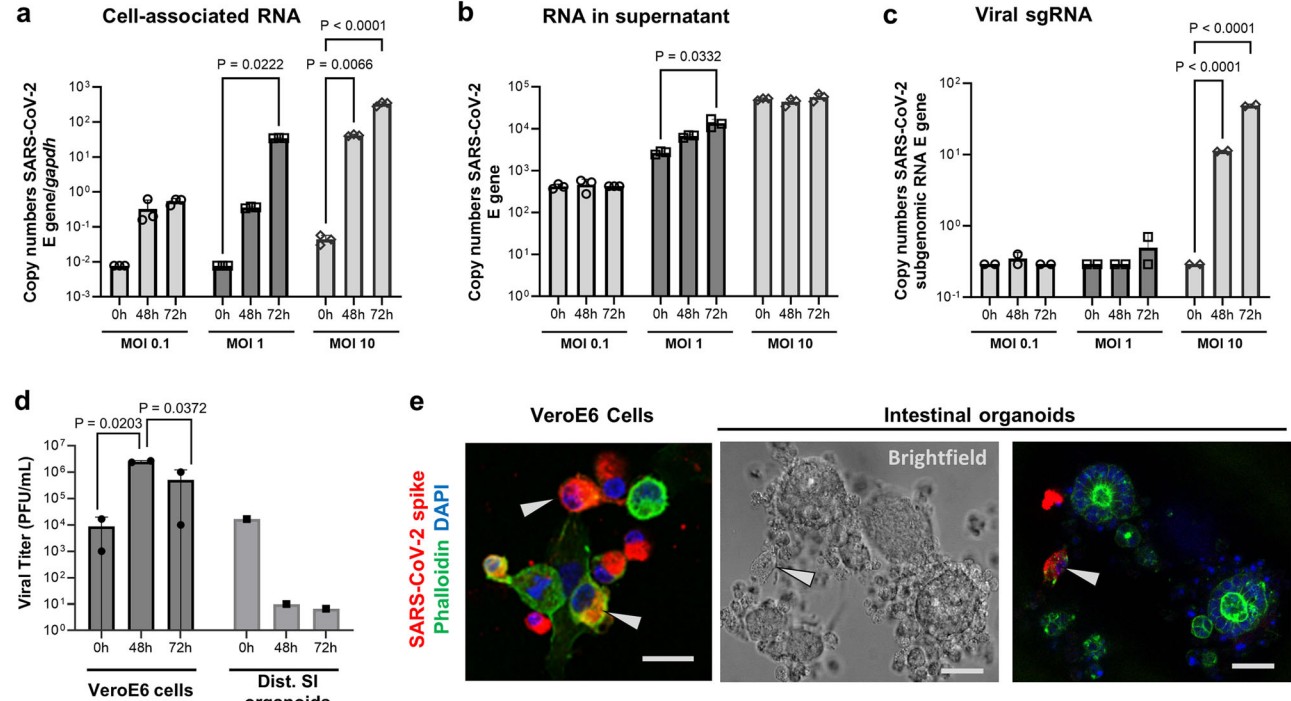

**Fig. 3 | Replication of SARS-CoV-2 in JFB intestinal organoids.** Dissociated JFB distal intestinal organoids (bat001, p6) were inoculated with SARS-CoV-2 (strain USA-WA1/2020) for 2 h or were mock-treated and then washed and re-embedded in Matrigel. At 48 and 72 h post-infection, RNA was extracted from (**a**) the organoids and (**b**) the culture supernatants, and replication of SARS-CoV-2 was analyzed by quantitative real-time PCR (qRT-PCR) for the envelope (E) gene using the standard curve method. Panels show data from one representative out of four independent experiments with *n* = 3 technical replicates as mean ± SD, analyzed by ANOVA with Dunnett's multiple comparisons test. **c** RNA extracted from the organoids was analyzed for viral sgRNA (E gene) using a leader-specific primer. One representative out of three independent experiments with *n* = 2 technical replicates, mean ± SD, analyzed by ANOVA with Dunnett's multiple comparisons test. **d** Supernatants from SARS-CoV-2 infected organoids (pooled from three replicates) or Vero E6 cells (duplicate wells) were analyzed by plaque assay for the presence of infectious SARS-CoV-2; representative of four experiments. **e** SARS-CoV-2 protein detection in isolated epithelial cells, but not in intact JFB intestinal organoids. Organoids or Vero E6 cells were fixed and permeabilized at 48 h post SARS-CoV-2 infection (MOI 10) and then were stained with DAPI (blue), phalloidin (green) and a monoclonal antibody to SARS-CoV-2 spike protein (red). Arrows point out cells containing SARS-CoV-2 spike protein. Data are representative of three independent experiments. Scale bar= 25 μm. Source data are provided as a Source Data file.

Fig. 5a, expression of *Ifna* was upregulated at 48 hpi with an MOI of 10, while an MOI of 1 caused significant upregulation of the gene at 72 hpi. Gene expression of *Ifnb* also was significantly increased with both MOIs at 48 hpi and remained elevated with the lower viral dose at 72 hpi (Fig. 5b). Type III IFNs are known to play a role in mucosal antiviral immunity and SARS-CoV-2 infection and also may have unique functions in bats[60–62]. However, the type III IFN loci in JFBs are poorly annotated[42,43], and we were unable to generate functional primers based on the published genome. Interestingly, organoid infection with SARS-CoV-2 at an MOI of 1 significantly increased expression of the proinflammatory cytokines *Tnf* and *Il6* at 48 hpi, and *Il6* remained elevated at 72 hpi (Fig. 5c, d). The above data suggest that JFB distal organoids exhibited an anti-viral and pro-inflammatory response to SARS-CoV-2 infection.

To determine whether active viral infection was responsible for the observed induction of antiviral and inflammatory genes, or whether gene expression was induced by unspecific activation of pattern recognition receptors, we also treated the JFB organoids with a panel of TLR agonists targeting TLR2, 3, 7, and 9 and with UV-inactivated SARS-CoV-2 for 48 h. Notably, stimulation with TLR2/1 and TLR3 agonists led to increased expression of interferon and inflammatory cytokines 6 h post inoculation (Supplementary Fig. 5). However, no significant upregulation of these genes was observed with any of the stimuli at 48 h (Fig. 5e–h). These observations suggest that active infection with SARS-CoV-2 is required for sustained upregulation of antiviral and inflammatory gene expression.

## Impact of SARS-CoV-2 infection on the JFB intestinal epithelial cell proteome

A quantitative proteomic workflow based on data-independent acquisition (DIA) mass spectrometry was used to perform a comprehensive analysis of the cellular responses of JFB organoids to SARS-CoV-2 infection. The DIA analysis of SARS-CoV-2-infected and mock-infected enteroids after 48 h yielded a total of 8321 proteins and protein isoforms, based on protein FASTA files retrieved from the *A. jamaicensis* reference genome[63,64]. Interestingly, all detected proteins were present in both experimental conditions. A comparative analysis of mock-infected and SARS-CoV-2 infected JFB organoids revealed 63 upregulated and 155 downregulated proteins, including isoforms, with a ≥ 2-fold change at *P* ≤ 0.05 (Fig. 6a and Supplementary Data 1). To better understand antiviral responses in the JFB intestine, we next compared the identified proteins to a comprehensive list of human interferon-stimulated genes (ISGs, Supplementary Data 2)[65]. Interestingly, 100 of all identified JFB proteins could tentatively be classified as ISGs. However, only one of the ISG proteins, ribonucleases P/MRP protein subunit POP1 (POP1), was significantly upregulated in response to SARS-CoV-2, while four ISG proteins (ERLEC1, CFB, ARMCX3 and ITIH2) were significantly downregulated (Fig. 6b). Overall, top upregulated proteins, based on fold change in abundance, were hepatocyte growth factor-like protein/macrophage stimulatory protein (HGFL/MST1), CUB domain-containing protein 1-like, acyl-CoA-binding domain-containing protein 5 (ACBD5), ketosamine-3-kinase (KT3K) and insulin-like growth factor 2 mRNA-binding protein 1 (IGF2BP1,

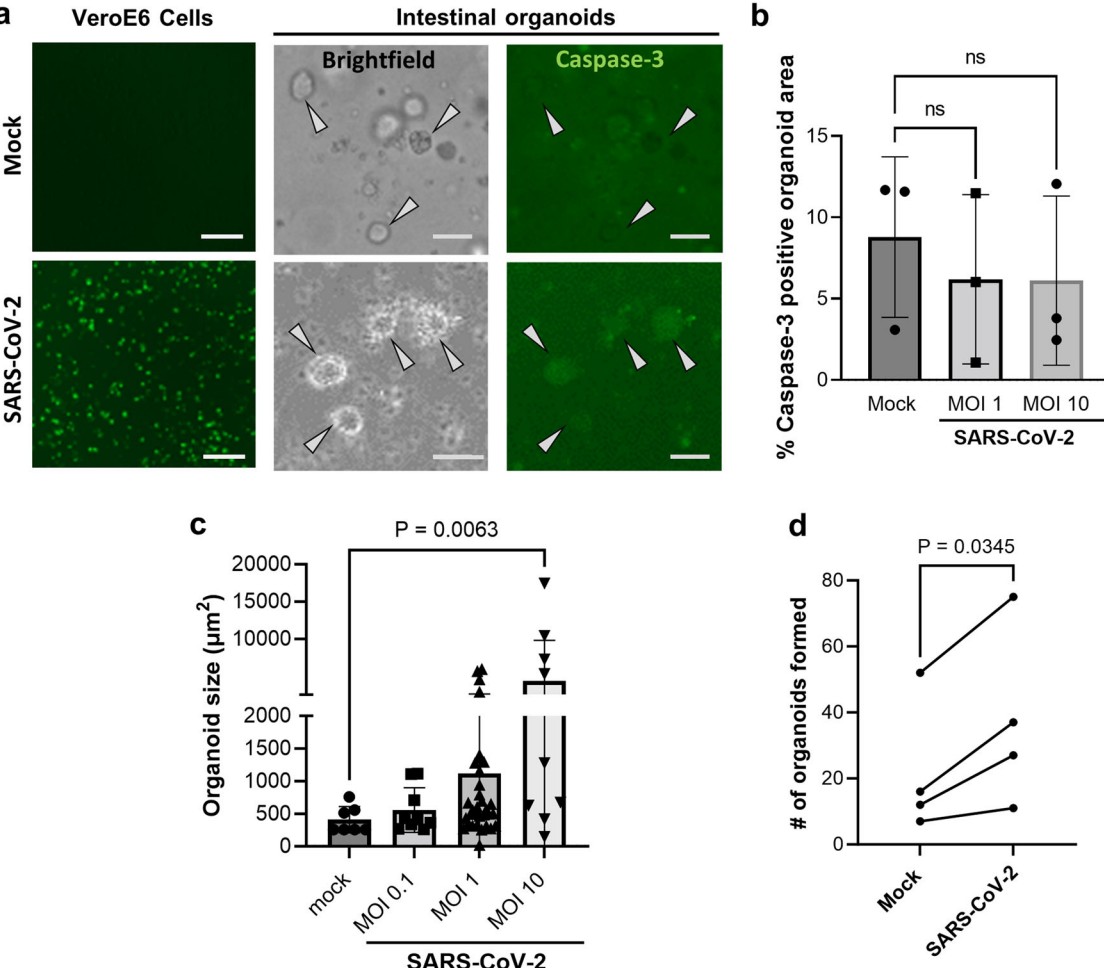

**Fig. 4 | Increased growth of JFB organoids infected with SARS-CoV-2.** Dissociated JFB distal intestinal organoids or Vero E6 cells were mock-inoculated or were infected with SARS-CoV-2 at an MOI of 1 or 10, as described above, with NucView® 488, a cell membrane-permeable fluorogenic caspase-3 reporter, added to the medium. **a** At 48 h post infection, the cells were imaged using fluorescence and phase contrast (brightfield) microscopy. Bat005, p7, scale bars = 50 μm, representative of four experiments. **b** ImageJ was used to quantitate NucView® fluorescence based on pixel counts in thresholded digital images of manually selected organoids. Individual data points, mean ± SD of one representative (bat002, p7) of four independent experiments with three technical replicates is shown, data were analyzed by ANOVA with Tukey's multiple comparison test. **c** Organoid size in SARS-CoV-2-infected organoid cultures after 48 h was determined on brightfield images using ImageJ. Individual data points, mean ± SD of one representative experiment (bat001, p6) of five independent experiments with three technical replicates is shown, data were analyzed by ANOVA with Tukey's multiple comparisons test. **d** Number of detected organoids in random brightfield images from mock-infected and SARS-CoV-2 infected JFB organoid cultures (MOI 10, 48 h). Pooled data from four independent experiments (bat001, p6; bat002, p7; bat003, p13; bat005, p5); analyzed using a paired 2-sided Student's *t* test. Source data are provided as a Source Data file.

Fig. 6c). Top down-regulated proteins included BTB/POZ domain-containing adapter for CUL3-mediated RhoA degradation protein 3 (KCTD10), CSC1-like protein 1 (TMEM63A), nuclear complex protein 3 homologue, histone H2A-β, and cell division complex protein 45 homologue (CDC45). Several of these proteins are involved in regulation of cell turnover and posttranslational modifications. We next performed Ingenuity Pathway Analysis (IPA) and Enrichr analysis[66] to assess more complex functional changes induced by SARS-CoV-2. IPA revealed acute phase response signaling, a key innate pathway triggered by infection and injury, as the most significantly regulated pathway, followed by Apelin liver signaling[67], which is involved in intestinal inflammation, repair, and wound healing (Fig. 6d). Top regulated cellular functions were cell assembly, organization, maintenance, movement, signaling and morphology (Fig. 6d). These findings suggest that SARS-CoV-2 triggers regenerative response pathways, consistent with the increased organoid size observed in the SARS-CoV-2-infected compared to mock-infected cultures. Similarly, Enrichr identified significant upregulation of pathways associated with cell viability and differentiation, such as PI3/AKT signaling and the longevity regulating pathway, along with signatures associated with intestinal epithelial infection and chemokine signaling when using the human 2021 KEGG pathways database (Fig. 6e). Importantly, Enrichr analysis also found multiple significant matches for protein signatures that were previously found to be upregulated in SARS-CoV-2 infection in Vero E6 cells[46,47,50], MA-104 cells[50], a human hepatocyte line[48], and human nasal epithelium[49] (Fig. 6f). Overall, the proteomics analysis points to the activation of innate inflammatory and regenerative pathways along with characteristic COVID-19 signatures upon SARS-CoV-2 infection of the JFB intestinal epithelium.

## Discussion

In this study, we established and characterized organoid cultures from the proximal and distal intestine and stomach of JFBs, a frugivorous species of New World bats, which have generally been underrepresented in immunological and virological studies[44,45]. Using this organoid model, we investigated the response of JFB intestinal epithelial cells to infection with SARS-CoV-2. Considering that JFBs are susceptible to MERS-CoV, Zika virus, and rabies virus[4,40,41], we

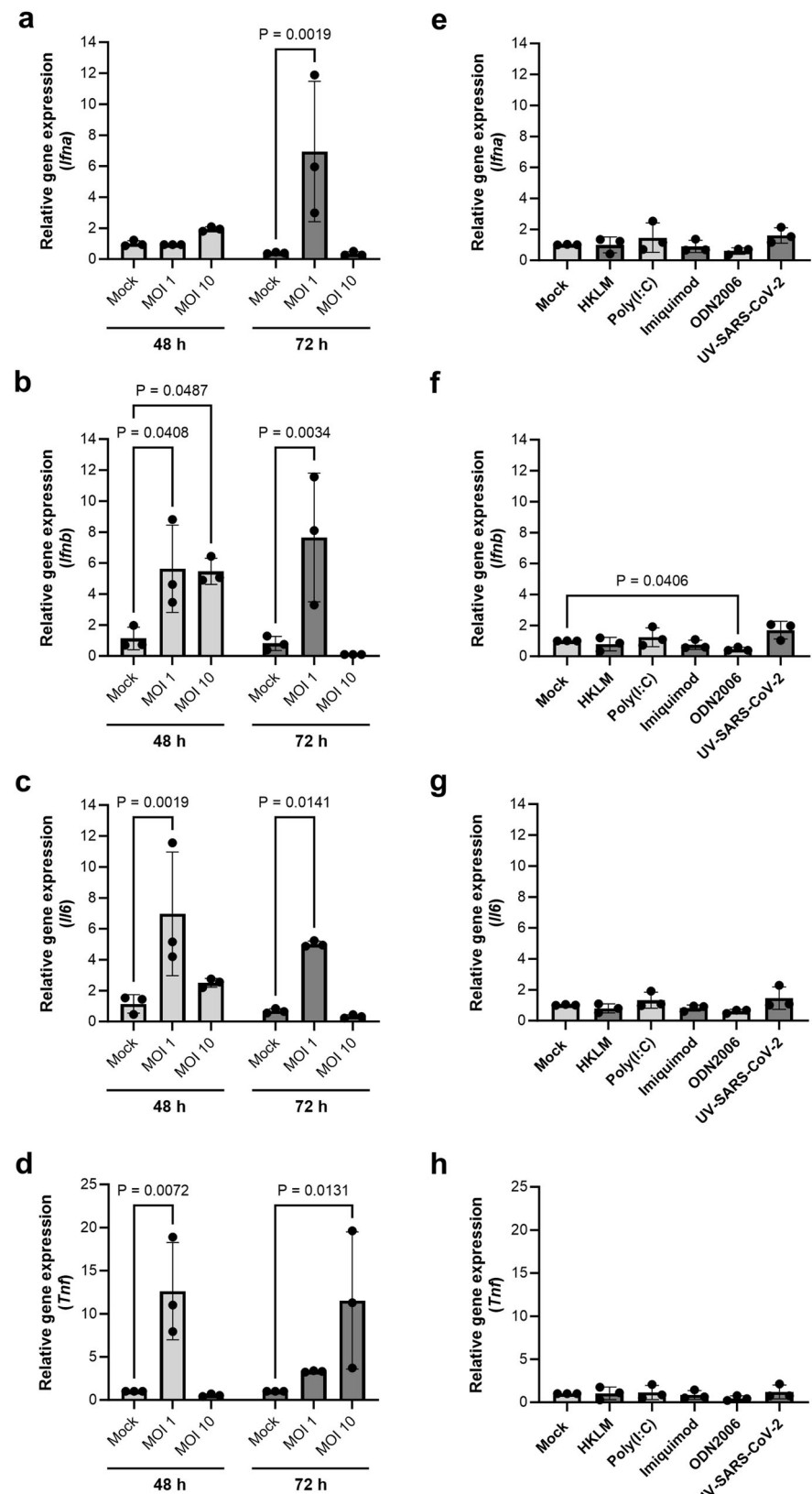

evaluated the susceptibility of the JFB distal intestinal organoids to SARS-CoV-2. We found evidence of limited, non-productive infection with induction of antiviral genes. Notably, JFBs are not thought to be natural carriers of SARS-CoV-2 or of related sarbecoviruses, and a recent preprint describing in vivo infection experiments in JFBs indicated that SARS-CoV-2 leads to an abortive infection of the intestine

without development of clinical disease[68]. Since SARS-CoV-2 is not adapted to JFBs, we assume that no host-specific viral immune evasion mechanisms have evolved, enabling activation of innate response pathways. Considering the vast number and associated genetic diversity of bat species, it is not surprising that SARS-CoV-2 infection experiments in other bat species have yielded conflicting results. In

**Fig. 5 | JFB distal intestinal organoids express antiviral and pro-inflammatory genes in response to infection with SARS-CoV-2. a–d** Dissociated JFB distal intestinal organoids (bat001, p6, three replicates) were infected with active SARS-CoV-2 at an MOI of 1 or 10. The unbound virus was washed off, and the cells were re-plated in Matrigel. After 48 or 72 h, the RNA was extracted from the cells to evaluate genes expression via quantitative real-time PCR (qRT-PCR). Data from one representative out of four independent experiments are shown as mean ± SD. **e–h** Organoids were treated with UV-inactivated SARS-CoV-2 at 10 μg/mL, or with a panel of TLR agonists (TLR2: heat-killed *L. monocytogenes*, HKLM; TLR3: low MW poly I: C; TLR7: imiquimod, TLR9: ODN2006) and then were analyzed by qRT-PCR 48 h after stimulation. Pooled data from three independent experiments (bat001, p6; bat004, p5; bat003, p5); mean ± SD are shown. Graphs show gene expression of (**a, e**) *Ifna* (IFN-α), (**b, f**) *Ifnb* (IFN-β), (**c, g**) *Il6* (IL-6) and (**d, h**) *Tnf* (TNF-α). All data were analyzed using the $2^{(-\Delta\Delta Ct)}$ method with *gapdh* as a housekeeping gene and are expressed as fold change relative to the mock-infected control. ANOVA with Dunnett's multiple comparisons test. Source data are provided as a Source Data file.

Egyptian fruit bats (*Rousettus aegyptiacus*), transient asymptomatic respiratory tract infection with viral replication in lung and trachea and oral and fecal shedding was achieved upon experimental SARS-CoV-2 inoculation[69]. Conversely, American big brown bats (*Eptesicus fuscus*) appeared resistant to infection with SARS-CoV-2[70]. Two independent studies on Brazilian free-tailed bats (*Tadarida brasiliensis)* by Bosco-Lauth[71] and Hall[72] found variable levels of susceptibility to SARS-CoV-2 infection in the absence of clinical signs. Likewise, intestinal organoids derived from two different bat species responded differently to SARS-CoV-2 infection. Organoids from Chinese horseshoe bats, where SARS-CoV-2-like virus has been detected[7], produced infectious SARS-CoV-2 virions at similar levels as human intestinal organoids[27]. In contrast, intestinal organoids from Leschenault's rousette bats (*Rousettus leschenaultii*) failed to support SARS-CoV-2 replication[25], and airway organoids from cave nectar bats (*Eonycteris spelaea*) allowed ACE2-dependent viral entry, but no productive viral replication[24]. In our hands, the JFB intestinal organoids consistently expressed low levels of ACE2 that could support viral entry. Interestingly, PCR analysis revealed a significant increase in viral and sgRNA in the JFB distal organoids at 48 and 72 hpi, which demonstrates initiation of viral replication in the organoids. SARS-CoV-2 genomes also were significantly increased in organoid culture supernatants, whereas plaque-forming units were detected but did not increase over time. Thus, it remains unclear whether any infectious or non-infectious virions were released. Using immunohistochemistry, we detected SARS-CoV-2 spike protein in individual cells, but not in morphologically intact JFB organoids. This observation may reflect shedding of viable virus-infected cells from the epithelial monolayer, as described for other viral infections[73]. Overall, our data suggest that JFB intestinal organoids support incomplete SARS-CoV-2 infection. A similar limited and incomplete replication of SARS-CoV-2 was also reported in cell lines from several different bat species, even after transduction with human ACE2, in a recent study by Aicher et al.[74]. However, the presence of sgRNA and of SARS-CoV-2 protein in some cells suggest that entry and replication of the virus did occur in the JFB organoids. This interpretation is consistent with a study by Yan et al. that predicted a moderate ability of SARS-CoV-2 to infect JFB cells based on the protein sequence of the SARS-CoV-2 receptor ACE2[75] and our observations of low ACE-2 gene expression in the JFB organoids. Loss of the furin cleavage site in the WA01 reference stain of SARS-CoV-2 also may have had an impact on the efficacy of infection[76]. Further experiments are needed to evaluate at which stage of the viral replication cycle SARS-CoV-2 replication stalls in the JFB organoid model and upon in vivo infection of JFBs[68]. Notably, many previous studies on viral infection in bats have relied solely on viral nucleic acids to measure infection[10–12,39,70,77]. Therefore, it is difficult to assess whether the failure to detect replication-competent virions was unique to our infection model.

Our results demonstrate that active SARS-CoV-2 virus induced a robust anti-viral immune response, with increased expression of IFN-α and IFN-β at 48 h after SARS-CoV-2 infection. This induction of interferons in response to viral infection was surprising, since it has been proposed that the interferon system in bats is constitutively active, based on studies in Australian black flying foxes (*P. alecto*)[28,78]. Conversely, potent interferon responses were detected in serotine bats

(*Eptesicus serotinus*) and David's myotis bat cells upon SARS-CoV-2 infection[74]. These observed differences point to species-specific immune system characteristics in bats, consistent with the high level of genetic diversity in the order Chiroptera, which comprises over 1400 species.

The lack of a cytopathic effect in SARS-CoV-2-infected JFB organoids was an intriguing observation. It has been shown that SARS-CoV-2 causes neither apoptotic nor necrotic cell death in the gastrointestinal tract of infected human patients. Whether SARS-CoV-2 impacts viability of organoid cultures is still a matter of debate. Lamers et al.[55] observed increased apoptotic cell death in human enteroids at 60 hpi, and Zhou et al.[27] state that both human and horseshoe bat enteroids developed a cytopathic effect after SARS-CoV-2 inoculation. Conversely, studies by Stanifer et al.[60] and Zang et al.[79] did not describe increased cell death in human SARS-CoV-2 infected enteroids. Data from several studies in bat cells suggest that heightened IFN responses in these cultures may prolong viral infection by limiting pathogen-induced cell death through induction of anti-apoptotic genes including BCL-2 and PMAIP1[39,80]. While we did not detect an upregulation in anti-apoptotic factors in our proteome screen of SARS-CoV-2-infected JFB organoids, we found a significant upregulation of pathways associated with cell growth and repair, including apelin liver signaling and wound healing signaling. These observations were consistent with the increase in organoid size and organoid formation that we detected by microscopic analysis and indicate activation of growth and repair pathways in response to SARS-CoV-2 infection. Taken together, our findings suggest that bat organoids activate protective repair pathways upon viral infection that may enable the bats to tolerate viral infection in the absence of tissue damage and associated clinical signs.

We used a DIA-based proteomic approach to gain deeper insights into the cellular responses induced in the SARS-CoV-2 infected JFB intestinal epithelium. Notably, only few studies in bats and other non-model organisms have included proteomics[81,82], although proteomics techniques can be particularly useful in species where few specific reagents are available[83]. We confirmed the identity of the organoids as small intestinal epithelial cells based on expression of key enterocyte markers. Consistent with the increased gene expression of pro-inflammatory cytokines in SARS-CoV-2-infected JFB organoids that we detected, inflammatory pathways including the acute phase response and chemokine signaling also were induced at the protein level. Conversely, although SARS-CoV-2 infection-induced expression of type I interferon transcripts in JFB organoids, no significant increase in ISGs was detected on the protein level. This lack of ISG regulation is inconsistent with proteomics results obtained in SARS-CoV-2-infected human Calu-3 cells, which showed a strong induction of the antiviral ISG signature[84], and may reflect a JFB-specific disconnect between transcriptional activation of interferons and downstream ISGs that warrants further studies. Alternatively, downregulation of ISG proteins may have been caused by active downregulation of antiviral ISG pathways by SARS-CoV-2 accessory proteins. Importantly, Enrichr analysis also revealed that some of the activated pathways matched those identified by other studies on SARS-CoV-2 infection. Notably, there are several limitations to the proteomics approach undertaken in our study. First, the non-targeted DIA approach may not be sensitive enough to identify strongly regulated targets with a low overall

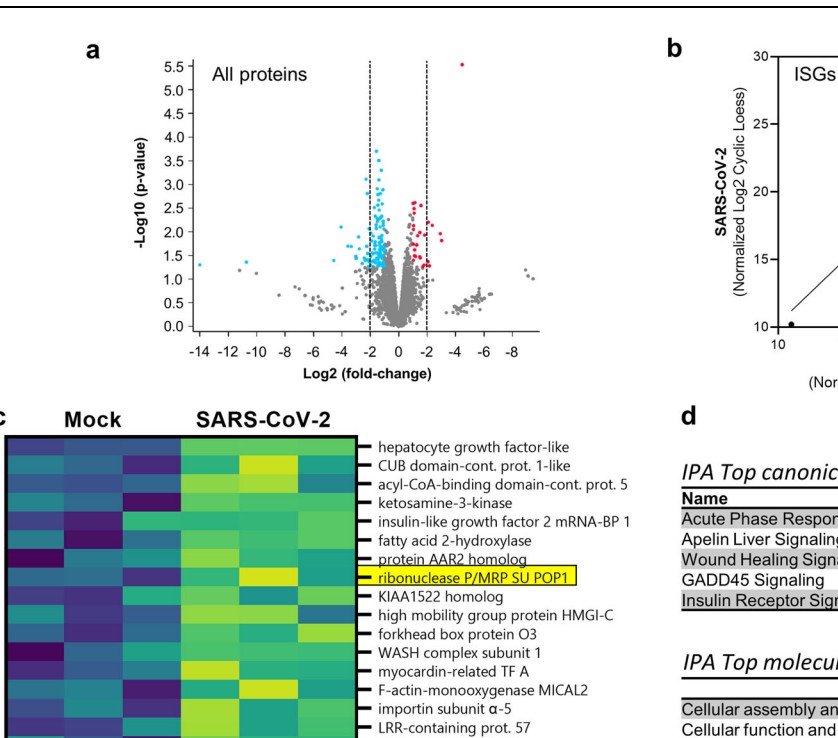

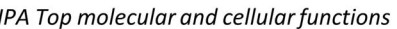

expression level[85]. Second, an annotated proteome of the JFB is currently not available and thus had to be inferred from the genome, which may lead to misidentified proteins. Lastly, pathway analysis was based on human databases, which again may miss JFB-specific signaling pathways.

Importantly, we successfully validated JFB organoids as an experimental tool and demonstrated that these JFB organoids can be maintained long-term without the need for bat-specific growth factors. Wnt, noggin and R-spondin are highly conserved in mammalian

species, with a high degree of sequence identity between mice and JFBs. The growth requirements for our JFB organoids are consistent with growth conditions previously described for Chinese horseshoe bats[27] and Rousettus bats[25]. Similar culture conditions also have been successfully used to culture intestinal organoids from cat, dog, cow, horse, pig and sheep[23]. We demonstrate that JFB organoids from stomach, proximal and distal intestine recapitulate the histology and morphology of the tissue of origin, with polarized columnar epithelial cells, mucus secretion, development of an intact epithelial barrier and

**Fig. 6 | Proteome analysis of SARS-CoV-2-infected JFB organoids at 48 h.** JFB distal SI organoids (bat003, p9) were infected with SARS-CoV-2 at an MOI of 10 or underwent mock treatment for 48 h and then were lysed and processed for data-independent acquisition (DIA) mass spectrometry. $N = 3$ replicates from one organoid line were analyzed. lmfit with empirical Bayes smoothing, a linear regression model from the Limma R package, was used with FDR correction for multiple tests. **a** Volcano plot showing all detected proteins and protein isoforms. Proteins with significantly increased or decreased expression ($\geq 2$-fold change; $P \leq 0.05$) are shown in red and blue. **b** Expression of interferon-stimulated genes (ISGs), identified based on OhAinle et al. (2018)[65], in mock-infected and SARS-CoV-2 infected JFB organoids. Proteins with significantly increased or decreased expression ($\geq 2$-fold change; $P \leq 0.05$) are shown in red and blue. **c** Heatmap showing relative change (Z-scores) of all 27 significantly upregulated proteins and of the top 30 downregulated proteins ($P \leq 0.05$). Data from triplicate cultures are shown. Protein function was determined using UniProtKB (*H. sapiens*). Significantly regulated ISGs are highlighted in yellow. **d** IPA analysis showing top regulated canonical signaling pathways (top) and molecular and cellular functions (bottom) activated in SARS-CoV-2 infected JFB organoids. Enrichr pathway analysis using (**e**) the 2021 human KEGG pathway database and (**f**) the 2021 COVID-19 related gene sets. Pathways were ranked based on combined score ranking. Source data are provided via the PRIDE partner repository with the dataset identifier PXD036016.

expression of tissue-specific genes and proteins. Thus, we have developed and validated a research tool that will allow experimental analysis of the physiology and function of the gastrointestinal epithelium of Jamaican fruit bats in future studies.

To summarize, we established and characterized JFB gastrointestinal organoids that recapitulated the organ-specific multicellular composition of JFB gastrointestinal tissue. We demonstrated SARS-CoV-2 sgRNA replication at a low efficiency in JFB distal intestinal organoids via qPCR but were unable to detect release of infectious virus. SARS-CoV-2 infection induced a robust upregulation of interferons and pro-inflammatory genes in the organoid cells. Moreover, SARS-CoV-2 infection of JFB organoids led to increased growth and activation of cellular regeneration and healing pathways, which might contribute to the improved viral tolerance in this bat species.

## Methods
### Ethics statement
The research presented here complies with all relevant ethical regulations and was approved by the Colorado State University Institutional Animal Care and Use Committee (IACUC) under protocol #1034.

### Tissue samples
Male and female Jamaican fruit bats (*Artibeus jamaicensis)* aged between one and eight years were maintained as an outbred breeding colony in an AAALAC-accredited facility at Colorado State University (CSU). For organoid derivation, five bats (4 males, bat001, 002, 004 and 005; and 1 female, bat 003) were euthanized by 5% isoflurane in $O_2$ followed by thoracotomy. The gastrointestinal tracts were harvested in RPMI-1640 medium and were shipped overnight on ice from CSU to Montana State University (MSU).

### Crypt and gland isolation methods
Bat tissues were processed immediately upon arrival or were cryopreserved and then thawed rapidly if needed[86]. To derive organoids, proximal intestinal and distal small intestinal tissues were washed in cold PBS and cut into ~1 mm pieces. The minced tissue was incubated in 15 mM EDTA in PBS supplemented with penicillin, streptomycin, and Fungizone (GE Healthcare Life Sciences) with gentle shaking for 10 min increments until crypts appeared in the supernatant. Large tissues pieces were removed by sedimentation. The supernatant containing the crypts was transferred into a new 50 mL tube and pelleted by centrifugation for 8 min at 150 $g$. Gastric tissues were digested for one hour at 37 °C using a digestion solution containing 5 U/mL collagenase type IV and 0.2 mg/mL DNAse (both Sigma-Aldrich), following our published protocols[87,88]. Recovered crypts/glands were resuspended in 10 μl of Matrigel and plated in 96-well plates. After the gel was polymerized, 200 μl of medium (Supplementary Table 1) was added, and the plates were incubated at 37 °C with 5% $CO_2$ for one week.

### Maintenance of JFB organoids
For passaging, the Matrigel patties containing organoids were digested for 3 min in TrypLE (Gibco) at 37 °C and pipetted up and down 50 times. The digested organoids were harvested by centrifugation for

5 min, 200 $g$ at 4 °C, then were resuspended in Matrigel and plated in a 24-well plate. After the gel had polymerized, 500 μl of 50% L-WRN-conditioned medium[51] (Supplementary Table 1) was added and the plate was incubated at 37 °C with 5% $CO_2$. The medium was changed every other day and the organoids were passaged every 5–7 days. In general, organoids at passages three to ten were used for experiments.

### Optimization of growth conditions
In addition to the basic growth medium, termed L-WRN medium, described above, we also tested a commercially available growth medium, IntestiCult™ (StemCell), a complex medium termed "colonoid medium" described by Tsai et al.[86], and analyzed medium supplementation with a number of different growth factors commonly used in organoid culture protocols (Supplementary Table 1). We prepared a medium with all available growth factors (L-WRN Plus) and then eliminated one reagent at a time from L-WRN Plus to determine the influence of the reagent on organoid growth. For this assay, the organoids were digested with TrypLE for 3 min and plated in a 96-well plate with the different media. Cell viability and proliferation were measured using the CellTiter-Glo luminescence assay (Promega).

### Microscopic analyses
Images of organoid cultures were captured using a Keyence BZ-X800 microscope with BZ-X800 Viewer software, v01.02.03.02 or an EVOS FL Auto System (ThermoFisher). Size and morphology of organoids was analyzed using OrganoSeg software[52] and phase contrast images. Measurements were performed on digital images by blinded investigators using ImageJ software. For histological analyses, organoids were recovered from the culture plates and treated with Histogel (ThermoFisher) prior to formalin fixation and paraffin embedding, following standard protocols. Slides were stained with hematoxylin/eosin and with Alcian Blue to visualize mucus production. TEM was performed as previously described for human organoids[87]. Briefly, organoids were fixed in 3% glutaraldehyde, processed for ultrathin sectioning, and then were imaged on a Zeiss LEO 912AB TEM.

### Immunofluorescence staining
To detect enterocyte markers and ACE2, we used antibodies to epithelial cytokeratin (1: 50, ThermoFisher, #4545; clone C11), villin (1: 100, Invitrogen, PIMA516408, clone SP145) and ACE2 (1: 100, R&D Systems, AF933, goat polyclonal) that have known reactivity across multiple different species. To detect SARS-CoV-2 protein in the organoid cultures, a monoclonal antibody to SARS-CoV-2 (11G10-F8) was generated in house, using a standard hybridoma protocol[89]. Briefly, mice were immunized with 10 μg UV-inactivated SARS-CoV-2 (USA-WA1/2020)[90] in Titermax adjuvant (Sigma) three times separated by at least two weeks. 11G10-F8 was then generated from a fusion of mouse splenocytes with SP2/0 cells. Mouse sera were screened for reactivity to the virus by ELISA. Clone 11G10-F8 recognizes the RBD region of the S1 subunit of the spike protein, as determined by ELISA, and was used at a concentration of 10 μg/mL. For immunofluorescence analysis, organoids were fixed with 4% PFA, permeabilized with 0.2% Triton X-100, and then treated with blocking buffer (DPBS with 10% FBS, 0.2%

Triton X-100, 0.1% BSA, and 0.05% Tween) overnight. After washing, samples were incubated with primary antibody for 2 h at room temperature. Then the secondary antibodies (anti-rabbit IgG AlexaFluor 555, 1:50, Southern Biotechnology, #4050-32; goat anti-mouse IgG (H+L) AlexaFluor 594, 1: 100, Invitrogen, A11005; or rat anti-mouse IgG1 eFluor660, 1: 100, eBiosciences, 50-112-4348), were added and incubated for 2 h at room temperature. The nuclei were stained with 5 μM DAPI (MP Biomedicals, 0215757405). Actin filaments were stained with ActinGreen 488 ReadyProbes reagent (Invitrogen, R37110). Stained organoids were imaged on an inverted SP5 Confocal Scanning Laser Microscope or an inverted DMI8 Stellaris (Leica) with LAS X software version 4.5.0 or earlier using a 20x objective (W 2010; Zeiss, Oberkochen, Germany). Z-stacks of 2-11 randomly selected organoids with intact morphology for each experiment and condition were recorded.

## Transepithelial resistance

To assess development of barrier function, organoids were dissociated and re-seeded on transwell inserts (Costar, Corning, 3 μm pore size) coated with collagen I. Transepithelial resistance was measured daily using a Voltohmmeter (EVOM2™, World Precision Instruments) and is expressed as $\Omega*cm^2$.

## SARS-CoV-2 infection of JFB organoids

Bat organoids were dissociated by incubation with 350 μL TrypLE to expose the apical and basolateral epithelial surface to the virus. Dissociated organoids were transferred to a BSL3 laboratory and then inoculated with SARS-CoV-2 (strain USA-WA1/2020, BEI Resources), at a multiplicity of infection (MOI) of 0.1, 1 and 10 for 2 h at 37 °C with frequent gentle agitation. Notably, the SARS-CoV-2 strain used was shown to have a defective furin cleavage site[91], but readily infected inducible pluripotent stem cell-derived human intestinal organoids in control experiments. Following incubation with the virus, organoids were collected into 500 μL DMEM and centrifuged at 200 g for 5 min to wash. Then cells were resuspended in 30 μL Matrigel and plated. After 10 min to allow gelation of the Matrigel, medium was added to the organoids. This medium was removed and fresh medium added to eliminate free viral particles. Then the plates were incubated at 37 °C for the indicated intervals. Infectious particles in culture supernatants were detected for each time point by plaque assay on Vero E6 cells, as previously described[90].

## Treatment of JFB organoids with TLR agonists and inactivated virus

To analyze the transcriptional response of JFB organoids to stimulation with pathogen-associated molecular patterns, organoids were trypsinized and then re-embedded into Matrigel in the presence of the following TLR agonists (Human TLR1-9 agonist kit, InvivoGen): TLR1, Pam3CSK4, 1 μg/mL; TLR2, heat-killed *Listeria monocytogenes* ($10^8$/mL); TLR3, low molecular weight poly I:C, 10 μg/mL; TLR7, imiquimod, 1 μg/mL; TLR9, ODN2006, 5 μM. Alternatively, organoids were treated with UV-inactivated SARS-CoV-2[90] (10 μg/mL). After 48 h, organoids were lysed in TRI Reagent (Sigma) and processed for RNA isolation and RT-PCR.

## Quantitative RT-PCR

To analyze gene expression and cell-associated viral RNA, RNA was extracted from organoids using the Direct-zol RNA Miniprep-Plus (Zymo Research). The RNA was converted to cDNA using iScript Reverse Transcription Super mix for RT-qPCR (BioRad). Primers for gastric and intestinal epithelial cell-specific genes and cytokines were designed using NCBI primer blast using the JFB genome (*Artibeus jamaicensis*, NCBI:txid9417,) and are listed in Supplementary Table 2. GAPDH was amplified as housekeeping gene in each PCR reaction. For each gene, a standard curve was created, and gene copy numbers for each gene of interest were normalized to the copy numbers of the

housekeeping gene, GAPDH. To quantify SARS-CoV-2 in the organoid supernatant, viral RNA was extracted from culture supernatants using the QIA®Amp Viral RNA Mini kit (Qiagen). Viral genomes were then quantified in a single-step RT-PCR reaction using primers and a TaqMan probe to the SARS-CoV-2 envelope (E) gene, as previously described[90], and the Quanta Bio ToughMix Master Mix. In addition, a forward primer to the leader sequence was used together with the reverse primer and probe to detect E gene sgRNA as described by Wölfel et al.[58]. An RNA standard curve generated from a T7 in vitro transcribed gBlock™ sequence (Integrated DNA Technologies) was used for normalization.

## Cell viability and organoid growth

To measure caspase-3 activity in SARS-CoV-2-infected organoids, NucView488 (Biotium) was added to the medium at 3 μM once the organoids were re-plated following incubation with the virus. For measuring caspase-3 activity, the organoids were imaged using Life Technologies EVOS FL Auto system with a 10x objective. The images were analyzed using ImageJ version 1.48 V and NucView positive pixels were counted automatically on the thresholded images. Brightfield images of the organoid cultures were used to measure organoid size for normalization of the NucView data and for assessment of organoid growth.

## Proteomics analyses

Triplicate samples of distal intestinal organoids were infected with SARS-CoV-2, MOI 10, for 48 h as described above and then were lysed in RIPA lysis buffer (25 mM Tris/Cl, 150 mM NaCl, 1% NP-40, 1% SDS, 1% protease inhibitor) by passing the samples through a 26.5 G needle 5 times on ice. Samples were stored at −80 °C until they were analyzed at the IDeA National Resource for Quantitative Proteomics. An Orbitrap Exploris 480 was used for data-independent acquisition (DIA) mass spectrometry with a 60 min gradient per sample and gas-phase fractionation to obtain comprehensive proteomic profiles of the organoids. Chromatogram libraries were constructed using Prosit[92], and proteins were identified and quantified using EncyclopeDIA, based on protein FASTA files retrieved from NCBI RefSeq for the Jamaican fruit bat (BioProject PRJNA673233)[63,64]. The mass spectrometry proteomics data have been deposited to the ProteomeXchange Consortium via the PRIDE[93] partner repository with the dataset identifier PXD036016. False discovery thresholds of 1% were applied. The ProteiNorm app was used to optimize data normalization[94], and Scaffold DIA version 3.3.1 (Proteome Software, Portland, OR) was used for visualization. The MS2 exclusive intensities were normalized using cyclic loess and linear models for microarray (limma), and lmfit with empirical Bayes smoothing was used for the analysis[95]. Proteins with an FDR-adjusted P-value ≤ 0.05 and an absolute fold change ≥2 were considered significant. Ingenuity Pathway Analysis (Qiagen)[96] and Enrichr[66] with combined score ranking (c=log(p) * z, where c = the combined score, P=Fisher exact test P-value, and z=z-score) were used to identify cellular signaling pathways. Relevant public databases queried by Enrichr were Human KEGG pathways 2021 (https://www.kegg.jp/) and COVID-19 Related Gene Sets 2021[97] (https://maayanlab.cloud/covid19/). To analyze the impact of SARS-CoV-2 infection on ISGs, proteins identified in the JFB organoids were compared to a comprehensive list of ISGs[65] using a Python script.

## Statistical analyses

All data, except the proteome data, were analyzed using GraphPad Prism 9.5.1 or earlier versions and are shown as individual data points with mean ± SD. Comparisons between two treatments were made using a 2-tailed Student's t test or the Kruskal–Wallis test for data that was not normally distributed, and comparisons between multiple treatments were made using one-or two-way ANOVA with Dunnett's or Tukey's post hoc tests. Each sample was analyzed one time.

**Reporting summary**

Further information on research design is available in the Nature Portfolio Reporting Summary linked to this article.

## Data availability

The mass spectrometry proteomics data have been deposited to the ProteomeXchange Consortium via the PRIDE[93] partner repository with under accession code PXD036016. The raw data underlying bar charts and scatter plots that support the findings of this study are provided in the Source Data file and in Figshare under accession code https://doi.org/10.6084/m9.figshare.23536797[98]. Raw imaging data are available from the corresponding author upon reasonable request. Relevant public databases queried by Enrichr are Human KEGG pathways 2021 (https://www.kegg.jp/) and COVID-19 Related Gene Sets 2021 (https://maayanlab.cloud/covid19/). Source data are provided with this paper.

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

## Acknowledgements

Funding from the National Institutes of Health (U01EB029242-02S1 to D.B., C.B.C., M.A.J., S.T.W., and J.N.W.; R21AI169536 to D.B.; and R01AI140442 to T.S.), the Montana State University Office of Research, Economic Development and Graduate Education (D.B., C.B.C.), the Montana Agricultural Experiment Station (D.B., M.A.J.), and the Kopriva Family Foundation (M.H.) is gratefully acknowledged. We greatly appreciate the support of Dr. Aga Rynda-Apple, Evelyn Benson, and Caylee Falvo for help with developing bat-specific experimental meth-ods, B. Tegner Jacobson for preparing a Python script to compare comprehensive protein lists, and Conner Killeen and Travis Van Leeuwen for performing OrganoSeg analyses. We also would like to thank Drs. Raina Plowright, Arinjay Banerjee, Vincent Munster, Emi DeWit, Steve Smith, and Alyssa Evans for helpful discussions about this study. The National Resource for Quantitative Proteomics is supported by NIH grant R24GM137786.

## Author contributions

M.H.: conceptualization, methodology, investigation, formal analysis, writing—original draft; T.A.S.: methodology, investigation, formal ana-lysis; J.F.H.: investigation, methodology, formal analysis; D.Snyder: investigation; K.N.L.: investigation, methodology; S.D.B.: investigation, methodology, formal analysis; S.G.M.: investigation, methodology; M.D.C.: investigation, methodology; D.S.: investigation; D.C.: metho-dology; A.R.: investigation; B.S.: investigation, visualization; A.K.: meth-odology, investigation, supervision; E.K.L.: resources, methodology; M.P.T.: resources, methodology, supervision; C.C.: conceptualization, funding acquisition, Supervision; J.N.W.: conceptualization, funding acquisition; S.W.: conceptualization, funding acquisition; T.S.: resour-ces; M.J.: conceptualization, funding acquisition, supervision; D.B.: conceptualization, funding acquisition, project administration, formal analysis, writing—original draft and revisions, supervision. All authors have reviewed the final draft of the manuscript.

## Competing interests

The authors declare no competing interests.
