## [Peer Review File · Nature Communications]

Antiviral responses in a Jamaican Fruit Bat intestinal organoid model of SARS-CoV-2 infectionREVIEWER COMMENTS

Reviewer #1 (Remarks to the Author):

Hashimi et al., Antiviral response mechanisms in the Jamaican Fruit bat intestinal organoid model of SARS-CoV-2 infection

The manuscript by Hashimi et al describes the generation and characterisation of intestinal organoids and infection with SARS-CoV-2. Organoids and other ex vivo platforms are being increasingly described across a range of species and provide a useful platform for studies of antiviral immunity. The manuscript is well written but I have a couple of comments/queries.

Introduction

P5, line 100: What species is the following sentence referring to and can the authors please provide a reference?

"...matched profiles found in other SARS-CoV-2 infection studies..."

Results

The characterisation of the organoids shows some differences in expression of markers between passages. Can the authors include the passage number used for the infectious work as this may have an influence on other gene and protein expression.

The results of plaque assays and immunohistochemistry provide evidence that the SARS-CoV-2 infection failed to produce infectious virus or infect intact organoids. It would be useful to see where ACE-2 is expressed on these cells and its expression level. The authors mention ACE-2 expression in the Discussion as unpublished observations. This data should be included in the manuscript.

The Introduction states that there are 5 IFN α genes in the JFB genome. This manuscript describes the expression of IFN α 4I. Were the primers used to detect this gene able to cross-react with the other 4 IFN α genes? Why was this one chosen for the analysis?

Reviewer #2 (Remarks to the Author):

This study presents a rare development and validation of an organoid model in a common bat reservoir host species, the Jamaican fruit bat, and uses this to assess the immune response in vitro to SARS-CoV-2. The results are compelling, particularly given the seemingly strong antiviral response observed in this species (compared to other studies). I do have several suggestions for improvement and some minor minor edits outlined below.

L45: I would remove Ebola virus from this list, as bats remain unconfirmed as reservoir hosts for this particular virus. Nipah virus would be a better example with experimental confirmation.

L47: I would suggest qualifying this statement slightly as "which is also thought to have its evolutionary origins in bats"; as the authors note in the following sentence, SARS-CoV-2 has not been identified in wild bat populations, only precursor viruses.

L51-56: It may also be worth noting another recent study on wild bat coronaviruses (Becker et al. 2022 *Frontiers Virology*), in which viruses were relatively common in rectal swabs and also showed little evidence of clinical disease (weak differential abundance of immune proteins via proteomics).

L69-75: The authors should also mention and cite recent work by Chan et al. 2022 (*Emerging Microbes & Infections*), which established organoids (airway) from cave nectar bats (*Eonycteris spelaea*).

L75: Another aspect of novelty that the authors should emphasize here is that all organoids (and

most in vitro resources) are in Old World bat species, mirroring more general biases in broader bat virus research (e.g., see a systematic analysis by one of the coauthors, Crowley et al. 2020 Vaccines, as well as a coronavirus-specific analysis by Cohen et al. 2022 bioRxiv). There remains an important need for more research effort (especially for in vitro resources) for New World bats.

L87: I would also provide the Latin name for JFB here.

L88: This is a bit semantic, but JFBs harbor rabies virus (rabies being the disease).

L89: It is also worth noting that JFBs commonly forage close to human settlements (e.g., backyard fruiting trees, crop fields), which could more readily facilitate virus spillovers.

L91: The authors mention recently annotated JFB genomes by Shaw et al. 2012 and Wang et al. 2020, but it is also worth noting another high-quality and recent annotation by Scheben et al. 2020 (bioRxiv). How similar are the Wang and Scheben annotations? Would using the latter possibly result in better transcriptomic and proteomic mapping?

L186: Does this change at all if another high-quality reference annotation is used? The authors may consider mapping to the Scheben et al. annotation for JPR, depending on the coverage of this vs the Wang et al. annotation.

L248: It may be worth qualifying here that no New World bat species are natural carriers of SARS-CoV-2 (or any SARS-like coronavirus), given that sarbecoviruses are only naturally found in particular taxa of Old World bats.

L256: The authors may also consider discussing recent experimental infections of Mexican free-tailed bats by SARS-CoV-2 (Bosco-Lauth et al. 2022 Viruses, Hall et al. 2023 Msphere), which even within one species have suggested differing results.

L284-288: One possibility the authors should consider in this interpretation (why JFB mount such a strong antiviral response to SARS-CoV-2) is that this subgenus of virus has closely coevolved with Old World bats (especially rhinolophids; Latinne et al. 2020 NatComm) and thus may represent a fairly novel viral challenge. Related work recently on naturally circulating alphacoronaviruses in vampire bats suggested very little proteomic response to infection (Becker et al. 2022 Frontiers Virology), more consistent with the paradigm mentioned earlier by authors.

L311: It could also be worth emphasizing the general lack of proteomic approaches in studies of reservoir host immunology (i.e., see Heck & Neely 2020 JPR for a general discussion as well as Hecht-Höger et al. 2020 MolEcology and Neely et al 2021 JPR for rare bat-specific examples, though using wild bat samples and not cell lines).

Reviewer #3 (Remarks to the Author):

The authors established and characterized Jamaican Fruit bat gastrointestinal organoids that recapitulated the organ-specific multicellular composition of the bat gastrointestinal tissue. They demonstrated significantly increased SARS-CoV-2 sgRNA in the bat distal intestinal organoids via qPCR assay but were unable to detect the release of infectious viruses. SARS-CoV-2 infection induced a robust upregulation of interferons and pro-inflammatory genes in the organoid cells. Moreover, SARS-CoV-2 infection of bat organoids led to the activation of cellular regeneration and healing pathways, which might contribute to the improved viral tolerance in this bat species. It is a great endeavor to establish bat organoids for studying the biology underlying bats carrying lots of coronaviruses in a disease-free manner. The strength of organoids over traditional cell lines is that organoids can faithfully simulate the tissues of origin. As such, a meticulous characterization of bat intestinal organoids for recapitulation of bat intestinal epithelium is a prerequisite for these bat organoids for modeling any biological processes. I'd suggest the author clarify the following issues.

- 1) Characterization of cell types in bat organoids (Fig.1B). the authors demonstrated the presence of goblet cells by Alcian Blue staining. However, enterocytes are the major cell type in organoids and the native intestinal epithelium. I'd suggest the authors characterize enterocytes within the organoids.
- 2) The authors showed the stability of the consecutive passage by organoid area and eccentricity in Fig. 1C. Cell-type gene expression profile might be a more favorable assessment to demonstrate the stability of bat intestinal organoids.
- 3) As far as I know, bats don't have large intestines. The authors may have to verify this in Jamaican fruit bats. If this is the case, authors may amend the terms to proximal and distal intestinal organoids, rather than small intestinal organoids.
- 4) The results of SARS-CoV-2 infection in bat organoids (Fig. 3) were not impressive. After MOI of 1 and 10 inoculation (which is quite high), intracellular viral RNA and those in the culture medium increased more or less over time. Increased sgRNA copy number was observed in 10 MOI infection. The conclusion "JFB organoids were susceptible to SARS-CoV-2 infection" in the abstract, line 31-32, might be insufficiently convincing.

AUTHOR RESPONSE TO REVIEWER COMMENTS

Reviewer #1

(1) Introduction

P5, line 100: What species is the following sentence referring to and can the authors please provide a reference?
“...matched profiles found in other SARS-CoV-2 infection studies...”

Author response:

We apologize for the lack of appropriate references, which we now have included in the introduction. The cell types that were used in the referenced studies on pathway analyses of SARS-CoV-2 infection have now been included in the results section, where the relevant data are described (lines 269-270).

(2) Results: The characterisation of the organoids shows some differences in expression of markers between passages. Can the authors include the passage number used for the infectious work as this may have an influence on other gene and protein expression.

Author response:

Thank you very much for this suggestion. To address this comment, we have included relevant organoid lines and passage numbers in all figure legends. We also have included new data, as suggested by reviewer #3, demonstrating relatively stable expression of enterocyte genes between passages 2 and 8. Notably, experiments were generally performed with organoids that were between passage three and ten, as now stated in the methods section (lines 426-427).

(3) The results of plaque assays and immunohistochemistry provide evidence that the SARS-CoV-2 infection failed to produce infectious virus or infect intact organoids. It would be useful to see where ACE-2 is expressed on these cells and its expression level. The authors mention ACE-2 expression in the Discussion as unpublished observations. This data should be included in the manuscript.

Author response:

We agree with the reviewer that ACE2 expression levels likely have a major impact on susceptibility to SARS-CoV-2 infection. We had previously performed qRT-PCR analysis and now have included ACE2 gene expression data in **Fig. 1D** and **E**. Importantly, *Ace2* was consistently expressed over multiple passages in the distal intestine, with cT values generally ranging between 27 and 32. Moreover, *Ace2* gene expression tended to be higher in the distal intestine than in the proximal intestine or the stomach. However, copy numbers relative to the housekeeping gene GAPDH and compared to other enterocyte genes were low.

We also now performed immunofluorescence analysis of JFB intestinal organoid whole mounts using a cross-reactive ACE2 antibody (R&D Systems, AF933), to determine ACE2 protein expression and cellular distribution. As now shown in **Fig. 2A**, we found weak expression of ACE2 on both the apical and basal cell surface. Conversely, ACE2 protein was not detected in the proteomics screen, which prioritizes highly expressed targets. Based on these results, we conclude that ACE2 is low but is expected to allow some degree of viral entry. This is now explained in the discussion (lines 301-303). Relevant data are described in the results, lines 140-147 and 153-154.

(4) The Introduction states that there are 5 IFN α genes in the JFB genome. This manuscript describes the expression of IFN α 4l. Were the primers used to detect this gene able to cross-react with the other 4 IFN α genes? Why was this one chosen for the analysis?

Author response:

We would like to thank the reviewer for drawing our attention to this issue. To address this question, we have used Primer BLAST to map the primers to the JFB genome. It appears that the *Artibeus jamaicensis* reference genome in the NCBI database has been updated since our last search and now shows only four protein-coding IFN- α genes: three variants of IFN- α -4-like and one variant of IFN- α -10-like. Importantly, the primers used in our study recognized all four IFN- α gene sequences. We have updated the primer list in the supplemental data, and we refer to *Ifna* rather than *Ifna4l* in the results and the figures. We also have revised the description of JFB IFN genes in the introduction (lines 102-103).

Reviewer #2:

(1) This study presents a rare development and validation of an organoid model in a common bat reservoir host species, the Jamaican fruit bat, and uses this to assess the immune response in vitro to SARS-CoV-2. The results are compelling, particularly given the seemingly strong antiviral response observed in this species (compared to other studies). I do have several suggestions for improvement and some minor minor edits outlined below.

Author response:

We would like to thank the reviewer for their positive feedback on our manuscript and for all the detailed suggestions. We greatly appreciate the reviewer's time, effort, and helpful input.

(2) L45: I would remove Ebola virus from this list, as bats remain unconfirmed as reservoir hosts for this particular virus. Nipah virus would be a better example with experimental confirmation.

Author response:

We appreciate this suggestion and have replaced Ebola virus with Nipah virus in the list, with an appropriate citation (line 48).

(3) L47: I would suggest qualifying this statement slightly as "which is also thought to have its evolutionary origins in bats"; as the authors note in the following sentence, SARS-CoV-2 has not been identified in wild bat populations, only precursor viruses.

Author response:

We have revised this sentence as recommended (line 50).

(4) L51-56: It may also be worth noting another recent study on wild bat coronaviruses (Becker et al. 2022 *Frontiers Virology*), in which viruses were relatively common in rectal swabs and also showed little evidence of clinical disease (weak differential abundance of immune proteins via proteomics).

Author response:

Thank you very much for drawing out attention to this interesting study, which we have now cited in the introduction (lines 59-61).

(4) L69-75: The authors should also mention and cite recent work by Chan et al. 2022 (*Emerging Microbes & Infections*), which established organoids (airway) from cave nectar bats (*Eonycteris spelaea*).

Author response:

As suggested by the reviewer, this important recent study is now cited in the introduction to our manuscript (lines 76-78) and in the discussion (lines 301-302). Notably, the airway organoids did not enable SARS-CoV-2 replication.

(5) L75: Another aspect of novelty that the authors should emphasize here is that all organoids (and most in vitro resources) are in Old World bat species, mirroring more general biases in broader bat virus research (e.g., see a systematic analysis by one of the coauthors, Crowley et al. 2020 *Vaccines*, as well as a coronavirus-specific analysis by Cohen et al. 2022 *bioRxiv*). There remains an important need for more research effort (especially for in vitro resources) for New World bats.

Author response:

This is an important point, and we appreciate the reviewer pointing out this issue. We have added information to the introduction (lines 108-109) and discussion (lines 278-279) highlighting the relevance of performing these analyses in Jamaican Fruit bats as a highly common New World bat species. We also have included the suggested citations for Crowley (*Vaccines* 2020) and Cohen (now published as a peer-reviewed in *Nature Microbiology* 2023) in our paper.

(6) L87: I would also provide the Latin name for JFB here.

Author response:

Thank you very much for drawing out attention to this oversight. We have added the Latin name *Artibeus jamaicensis* to the title, the abstract (line 30) and the introduction (line 96).

(7) L88: This is a bit semantic, but JFBs harbor rabies virus (rabies being the disease).

Author response:

We have corrected this error for rabies virus and West Nile virus (line 97).

(8) L89: It is also worth noting that JFBs commonly forage close to human settlements (e.g., backyard fruiting trees, crop fields), which could more readily facilitate virus spillovers.

Author response:

We agree with the reviewer that overlapping habitats of human and JFBs with an associated risk for disease spillover provides additional relevance to our investigations. We have added a sentence to the introduction (line 99) to convey this point.

(9) L91: The authors mention recently annotated JFB genomes by Shaw et al. 2012 and Wang et al. 2020, but it is also worth noting another high-quality and recent annotation by Scheben et al. 2020 (bioRxiv). How similar are the Wang and Scheben annotations? Would using the latter possibly result in better transcriptomic and proteomic mapping?

Author response:

We appreciate the reviewer's suggestion. However, since the paper associated with the mentioned assembly has not yet been peer-reviewed, it is difficult to assess the quality of this assembly, and we do not think that we have the expertise to use the assembly for building primers for our bats. As the pre-print mentions, immune genes are especially difficult to align...

(10) L186: Does this change at all if another high-quality reference annotation is used? The authors may consider mapping to the Scheben et al. annotation for JPR, depending on the coverage of this vs the Wang et al. annotation.

Author response:

As mentioned above, the pre-print by Scheben et al., and the associated genome assemblies have not yet undergone peer review. Therefore, we did not believe that there was a solid rationale for us to undertake these additional analyses.

(11) L248: It may be worth qualifying here that no New World bat species are natural carriers of SARS-CoV-2 (or any SARS-like coronavirus), given that sarbecoviruses are only naturally found in particular taxa of Old World bats.

Author response:

As suggested by the reviewer, we have added this information to the discussion (line 283).

(12) L256: The authors may also consider discussing recent experimental infections of Mexican free-tailed bats by SARS-CoV-2 (Bosco-Lauth et al. 2022 *Viruses*, Hall et al. 2023 *Mosphere*), which even within one species have suggested differing results.

Author response:

We greatly appreciate this suggestion and have included a discussion of the infection data from *Tadarida*

brasiliensis in our manuscript (lines 294-296). We also have included data from a recent pre-print that describes abortive *in vivo* infection of Jamaican fruit bats with SARS-CoV-2 (Burke et al., bioRxiv, 2023.2002.2013.528205; lines 284-286).

(13) L284-288: One possibility the authors should consider in this interpretation (why JFB mount such a strong antiviral response to SARS-CoV-2) is that this subgenus of virus has closely coevolved with Old World bats (especially rhinolophids; Latinne et al. 2020 NatComm) and thus may represent a fairly novel viral challenge. Related work recently on naturally circulating alphacoronaviruses in vampire bats suggested very little proteomic response to infection (Becker et al. 2022 Frontiers Virology), more consistent with the paradigm mentioned earlier by authors.

Author response:

The fact that Jamaican fruit bats would not normally be exposed to SARS-CoV-2 and related viruses is an important consideration that we now discuss in more detail in the manuscript, since this is a complex issue. Notably, the rapid type-I IFN response of the organoids / gut epithelium to SARS-CoV-2 (as opposed to *in vivo* infection of bats) would be mediated by broadly reactive viral pattern recognition receptors that are expected to respond to any virus or viral PAMP. However, we agree with the reviewer that viral adaptation might involve host-specific immune evasion mechanism of the virus that could downregulate innate response pathways. Such viral adaptation would not have occurred for SARS-CoV-2 in JFBs. Therefore, we hypothesize that there may be a lack of viral adaptation to the host rather than a lack of active tolerance development due to a novel antigenic challenge. This is now explained in more detail in the paper (lines 286-288).

(14) L311: It could also be worth emphasizing the general lack of proteomic approaches in studies of reservoir host immunology (i.e., see Heck & Neely 2020 JPR for a general discussion as well as Hecht-Höger et al. 2020 MolEcology and Neely et al 2021 JPR for rare bat-specific examples, though using wild bat samples and not cell lines).

Author response:

We greatly appreciate the reviewer recognizing the novelty of our approach. References to previous proteome investigations in bats have been included in the discussion (lines 354-356).

Reviewer #3:

The authors established and characterized Jamaican Fruit bat gastrointestinal organoids that recapitulated the organ-specific multicellular composition of the bat gastrointestinal tissue. They demonstrated significantly increased SARS-CoV-2 sgRNA in the bat distal intestinal organoids via qPCR assay but were unable to detect the release of infectious viruses. SARS-CoV-2 infection induced a robust upregulation of interferons and pro-inflammatory genes in the organoid cells. Moreover, SARS-CoV-2 infection of bat organoids led to the activation of cellular regeneration and healing pathways, which might contribute to the improved viral tolerance in this bat species. It is a great endeavor to establish bat organoids for studying the biology underlying bats carrying lots of coronaviruses in a disease-free manner. The strength of organoids over traditional cell lines is that organoids can faithfully simulate the tissues of origin. As such, a meticulous characterization of bat intestinal organoids for recapitulation of bat intestinal epithelium is a prerequisite for these bat organoids for modeling any biological processes. I'd suggest the author clarify the following issues.

1) Characterization of cell types in bat organoids (Fig.1B). the authors demonstrated the presence of goblet cells by Alcian Blue staining. However, enterocytes are the major cell type in organoids and the native intestinal epithelium. I'd suggest the authors characterize enterocytes within the organoids.

Author response:

We agree with the reviewer that a detailed characterization of the organoid model system is important. In the previous manuscript version, we had presented histological data (H&E), proteomics profiling, gene expression data and immunofluorescence detection of enterocyte markers and morphology. To address the reviewer's concerns, we have now included additional data to demonstrate that the JFB intestinal organoid cells have an enterocyte phenotype. Specifically, we have used cross-reactive antibodies to perform multi-color immunofluorescence analysis of the organoids (**new Fig. 2A**) and now show staining for villin and phalloidin, epithelial cytokeratin and phalloidin, and ACE2 and phalloidin. Notably, JFB organoids had a physiological inside-in conformation, with luminal villin, phalloidin, and ACE2 staining, and cytoplasmic expression of epithelial cytokeratin. In particular, the detection of villin by gene expression assay,

immunohistochemistry and proteomics supports the hypothesis that JFB organoids are largely composed of enterocytes. In addition, we have now also included transmission electron micrographs showing characteristic microvilli on the apical cell surface as well as the formation of apical junctional complexes as additional evidence of a differentiated enterocyte phenotype (**new Fig. 2B**). All characterization data are now compiled in Fig. 2 for increased clarity. The new data are described in the results, lines 148-157.

2) The authors showed the stability of the consecutive passage by organoid area and eccentricity in Fig. 1C. Cell-type gene expression profile might be a more favorable assessment to demonstrate the stability of bat intestinal organoids.

Author response:

To address the reviewer's suggestion, we have performed gene expression analysis of two JFB organoid lines over seven consecutive passages. As now shown in **Fig. 1E**, expression of Muc2, Villin, Cdx2, Ace2 and pepsinogen C were largely stable between passages two and eight. For experiments, we generally used organoids between passages three and ten, as now stated in the methods section.

3) As far as I know, bats don't have large intestines. The authors may have to verify this in Jamaican fruit bats. If this is the case, authors may amend the terms to proximal and distal intestinal organoids, rather than small intestinal organoids.

Author response:

This is an interesting point that we have been unable to resolve. A recent paper by Silva et al. (*Acta Chiropterologica* 2020) shows histological images of gut tissue with large intestinal morphology for *Artibeus planirostris*, and a publication on MERS-CoV infection in *Artibeus jamaicensis* by Munster et al. (*Sci Rep* 2016) references the colon as a sampling site. However, we were unable to find published histological images of JFB large intestine. In our hands, the most distal portions of the gut that we were able to collect from JFBs still had morphological characteristics of the small intestine, with prominent villi. Therefore, we agree with the reviewer that the terms "proximal intestine" and "distal intestine" are more appropriate. Accordingly, we have changed the terminology throughout the manuscript.

4) The results of SARS-CoV-2 infection in bat organoids (Fig. 3) were not impressive. After MOI of 1 and 10 inoculation (which is quite high), intracellular viral RNA and those in the culture medium increased more or less over time. Increased sgRNA copy number was observed in 10 MOI infection. The conclusion "JFB organoids were susceptible to SARS-CoV-2 infection" in the abstract, line 31-32, might be insufficiently convincing.

Author response:

We agree with the reviewer that viral replication in the organoids was not strong. While we observed a significant increase in cell-associated and free viral RNA and in subgenomic RNA that indicates viral replication, increase over baseline was relatively low, and we were unable to detect release of virions that could infect VeroE6 cells. We have described these somewhat contradictory findings in a more detailed manner in the results (line 184-185) and discussion (lines 302-322). For the abstract, we have now reworded the statement on viral replication to "JFB organoids support only limited viral replication" (lines 33-34) to represent our observations more accurately.

REVIEWER COMMENTS

Reviewer #3 (Remarks to the Author):

The manuscript is improved substantially after revision. However, there are still some issues and errors.

1. I've mentioned in the initial comments that bats have no large intestines. This is not accurate. I intended to say bats may have no appendix. I apologize for the inconsistency. The authors are suggested to read an earlier paper (Int. J. Morphol., 26(3):591-602, 2008. Comparative Intestinal Histomorphology of Five Species of Phyllostomid Bats). I would also suggest the authors clarify whether the bats (Jamaican Fruit bats) have an appendix or not, and seek assistance from a pathologist to ascertain the nature of intestinal tissues for organoid derivation.

2. Line 231-132. As far as I know, goblet cells are an essential cell type in the native intestinal epithelium of both small and large intestines.

3. Line 153. "Inside-in" conformation is confusing, should be "apical-in". 3D organoids maintained in matrigel show an "apical-in" polarity invariably (Viruses. 2023 May 14;15(5):1166. Cell Rep. 2019, 26, 2509–2520.e4.), which is not related to any physiology.

4. Line 152. Actin is not exclusively present on the apical side of organoids. It is distributed within the cytosol, attached to the cell membrane and enriched on the apical surface.

Reviewer #4 (Remarks to the Author):

This study aims to determine the antiviral responses elicited by the GI tract organoids derived from the Jamaican Fruit bat (JFB, *Artibeus jamaicensis*) challenged with SARS-CoV-2, representing the first study of its kind in a new world bat species. Antiviral responses in bat organoids were determined by qRT-PCR and proteomics (data independent acquisition mass spectrometry – uniquely applied to bat organoids). Virus replication was determined by qRT-PCR and the production of infectious virus present in culture supernatant was assessed by plaque assay in VeroE6 cells. Intriguingly, while there was evidence of viral replication, as determined by detection of a modest increase in levels of SARS-CoV-2 nucleic acid, no infectious virus was detected in the culture supernatant. Challenge of organoids with SARS-CoV-2 elicited an increase in the expression of type 1 IFN as well as pro-inflammatory cytokines and chemokines at 48 h, which was not observed with TLR agonists at the same time point, indicating active virus replication is responsible for sustained antiviral and pro-inflammatory responses. Unlike *Pteropus alecto*, these bat tissues do not appear to demonstrate constitutive expression of type 1 IFN. Analysis of the proteome of SARS-CoV-2 infected organoids revealed that only one ISG, ribonuclease P/MRP protein subunit POP1 (POP1) was significantly upregulated in response to SARS-CoV-2 infection. However, further mechanistic studies to determine the role of this ISG in lack of production of infectious virus was not pursued? Interestingly, SARS-Cov-2 infection led to an increase in organoid size and no evidence of cytopathic effects. Quantitative proteomics analysis suggests activation of innate inflammatory and regenerative pathways upregulated in organoids challenged with SARS-Cov-2. This study reports the generation of GI organoids from JFB and provides novel insights into the interplay between SARS-CoV-2 and the innate immune response in the Jamaican Fruit bat GI organoids and will be of interest to researchers (virologists, immunologists, biologists) studying why bats harbour pathogens without showing clinical signs of disease.

Specific comments (lines refer to marked up version of manuscript):

1. Abstract. Lines 33-34. Suggest adding the following underlined text "...indicating that JFB organoids support only limited viral replication but not viral reproduction"

2. Abstract. Lines 36 -37. "gene expression of type I interferon and inflammatory cytokines was induced in response to SARS-CoV-2 but not in response to TLR agonists". This statement is a little misleading since the organoids were responsive to some of the TLR agonists tested with

upregulation of gene expression observed at 6 hours – but this effect was not prolonged i.e not observed at 48 – 72 hours?

Suggest changing to “gene expression of type I interferon and inflammatory cytokines was induced in response to SARS-CoV-2 replication”

3. Line 89 –it would be appropriate to cite the original paper that first reported constitutive expression of IFNalpha in Pteropus alecto i.e - Zhou et al 2016 PNAS doi: 10.1073/pnas.1518240113.

4. Line 108 The authors emphasise in the introduction that their data is contrary to the “always on” paradigm for antiviral interferon responses to bats. However, it has already been reported in the literature that not all bat species display constitutive expression of antiviral factors (i.e. type 1 IFN in Egyptian fruit bats , Pavlovich et al 2018 Cell; doi: 10.1016/j.cell.2018.03.070.). This should be acknowledged in the introduction and cited.

5. Line 111 Introduction. The authors state that their findings is contrary to the “Always on” paradigm for antiviral interferon responses in bats. See point 4. The authors should temper this statement i.e. “responses in specific bat species”

6. Line 112 – not clear what is meant by “active SARS-CoV-2” – do they mean “infectious SARS-CoV-2”?

7. Line 307. The authors state that SARS-Cov-2 genomes were significantly increased in organoid culture supernatant. Are these genomes virion-associated which would suggest the egress of noninfectious virus particles?

8. Fig 2D. Figure legend – define error bars in the graph in the right and the number of independent assays from which these data were derived, which is also needed to justify the statistical analysis.

9. Line 323 – the authors state that studies are needed to determine whether JFBs are permissive to SARS-CoV-2 infection in vivo, yet newly added text (lines 285 – 287) states that a recent preprint describing in vivo infection experiments in JFBs indicates that it leads to an abortive infection of the intestine without development of clinical disease (ref 66). Please resolve this apparent contradiction.

10. “data is not shown” appears in two places (line 137 and line 186).

REVIEWER COMMENTS

Reviewer #3

1. I've mentioned in the initial comments that bats have no large intestines. This is not accurate. I intended to say bats may have no appendix. I apologize for the inconsistency. The authors are suggested to read an earlier paper (Int. J. Morphol., 26(3):591-602, 2008. Comparative Intestinal Histomorphology of Five Species of Phyllostomid Bats). I would also suggest the authors clarify whether the bats (Jamaican Fruit bats) have an appendix or not, and seek assistance from a pathologist to ascertain the nature of intestinal tissues for organoid derivation.

Author response:

This is an interesting question, since bats are relatively long-lived, and the presence of an appendix has been associated with longevity (Collard et al. 2021, J. of Anatomy). We consulted with a collaborating veterinary pathologist, Dr. Steve Smith, who confirmed that both the proximal and the distal intestinal organoid lines were derived from tissue that had the characteristic morphology of the small intestine, since distinct villi were present and goblet cells were relatively rare (Fig. 1B and Supplemental Figure 3B). This is now clearly stated in the results section of the manuscript (Lines 125-128) Villi were shorter in the distal compared to the proximal intestine, consistent with the observations by Gadhela-Alves in the publication kindly recommended by the reviewer.

Conversely, both the appendix and the cecum are part of the large intestine, which lacks villi. During tissue harvest, we were unable to identify any pouch-like structures extending from the intestinal canal that might represent a cecum (with or without appendix). Consistent with our observations, an additional literature search performed to address this issue found multiple reports that state that most fruit-eating bats including *Artibeus* spp. lack a cecum and appendix (Klite, J. Bacteriology 1965; Tedman and Hall, Aust. J. Zool 1985; Silva et al., Acta Chiropterologica 2020).

2. Line 231-132. As far as I know, goblet cells are an essential cell type in the native intestinal epithelium of both small and large intestines.

Author response:

The reviewer is correct that mucus-secreting goblet cells are present throughout the intestine. We have revised this statement to explain more clearly that these cells are more common in the distal parts of the intestine than in the proximal parts and that this characteristic difference is maintained in the organoids (lines 136-138).

3. Line 153. "Inside-in" conformation is confusing, should be "apical-in". 3D organoids maintained in matrigel show an "apical-in" polarity invariably (Viruses. 2023 May 14;15(5):1166. Cell Rep. 2019, 26, 2509–2520.e4.), which is not related to any physiology.

Author response:

We apologize for the use of lab jargon and have changed this statement to "The majority of organoids had a typical apical-in conformation" (line 158), consistent with the terminology introduced by Co et al (Cell Rep. 2019 and Nat. Prot. 2021).

4. Line 152. Actin is not exclusively present on the apical side of organoids. It is distributed within the cytosol, attached to the cell membrane and enriched on the apical surface.

Author response:

We agree with the reviewer that actin as a major cytoskeletal protein is distributed throughout the entire cell. However, our microscopic data do demonstrate strong phalloidin binding on the apical side of the cell, consistent with increased density of actin, which is typical for formation of microvilli and a terminal actin web in enterocytes. We have revised the statement to make this clearer (lines 157-161).

Reviewer #4:

1. Abstract. Lines 33-34. Suggest adding the following underlined text “...indicating that JFB organoids support only limited viral replication but not viral reproduction”

Author response:

We appreciate the reviewer’s suggestion and have revised the abstract accordingly (line 34).

2. Abstract. Lines 36 -37. “gene expression of type I interferon and inflammatory cytokines was induced in response to SARS-CoV-2 but not in response to TLR agonists”. This statement is a little misleading since the organoids were responsive to some of the TLR agonists tested with upregulation of gene expression observed at 6 hours – but this effect was not prolonged i.e not observed at 48 – 72 hours? Suggest changing to “gene expression of type I interferon and inflammatory cytokines was induced in response to SARS-CoV-2 replication”

Author response:

We agree with the reviewer that this brief statement in the abstract is misleading, since there appears to be a difference in the kinetics of the interferon response to TLR agonists vs. infectious SARS-CoV-2. The key point we were trying to make here is that the bat organoids mount both an interferon and an inflammatory cytokine response upon infection with SARS-CoV-2. We agree that it is not necessary to mention the TLR agonists in the abstract and have changed the sentence as suggested (lines 34-36).

3. Line 89 –it would be appropriate to cite the original paper that first reported constitutive expression of IFNalpha in Pteropus alecto i.e - Zhou et al 2016 PNAS doi: 10.1073/pnas.1518240113.

Author response:

We appreciate this suggestion and have included the appropriate reference in lines 85 and 87,

4. Line 108 The authors emphasize in the introduction that their data is contrary to the “always on” paradigm for antiviral interferon responses to bats. However, it has already been reported in the literature that not all bat species display constitutive expression of antiviral factors (i.e. type 1 IFN in Egyptian fruit bats , Pavlovich et al 2018 Cell; doi: 10.1016/j.cell.2018.03.070.). This should be acknowledged in the introduction and cited.

Author response:

The reviewer is correct that the “always-on” paradigm for interferon for bats does not always apply. Rather, multiple unique mechanism related to IFN activation and signaling have been identified in different bat species, as described in lines 85-99. We have now also cited the study by Pavlovich, as recommended by the reviewer (line 95), which shows that type I IFN in Egyptian fruit bats is low at baseline, but inducible upon viral infection. We have reworded some of the other statements in the paper, so that revised manuscript now presents a more balanced view about this issue (see lines 85-99, 114-116, 337).

5. Line 111 Introduction. The authors state that their findings is contrary to the “Always on” paradigm for antiviral interferon responses in bats. See point 4. The authors should temper this statement i.e. “responses in specific bat species”

Author response:

To address the reviewer’s concern, we have deleted the statement about the “always-on paradigm” from this sentence, since it is not really necessary to mention this controversial hypothesis again here (lines 114-115). We also have tempered a statement about interferon activation in the discussion (line 337).

6. Line 112 – not clear what is meant by “active SARS-CoV-2” – do they mean “infectious SARS-CoV-2”?

Author response:

The reviewer is correct. This statement was referring to infectious virus. We have revised this sentence (line 116).

7. Line 307. The authors state that SARS-Cov-2 genomes were significantly increased in organoid culture supernatant. Are these genomes virion-associated which would suggest the egress of noninfectious virus particles?

Author response:

Whether any infectious or non-infectious virions are produced in the JFB organoids is an intriguing question that we unfortunately have not been able to resolve. In addition to the increase in cell-associated and cell-free viral genomes, it appears that spike protein was produced in some organoid cells (Fig. 3E), but plaque assays in Vero cells did not show an increase in infectious virions that exceeded the original inoculum. These results could be explained by at least three possible scenarios: (1) the RNA in the culture supernatants could have been released from dying cells independent of viral particles, with no virion release; (2) low-level production of infectious virions by the organoids might have been masked by the leftover inoculum in the cultures; or (3) as suggested by the reviewer, non-infectious viral particles could have been released. To explain this uncertainty, we have revised the manuscript as follows: “SARS-CoV-2 genomes also were significantly increased in organoid culture supernatants, whereas plaque-forming units were detected but did not increase over time. Thus, it remains unclear whether any infectious or non-infectious virions were released.” (lines 312-314).

8. Fig 2D. Figure legend – define error bars in the graph in the right and the number of independent assays from which these data were derived, which is also needed to justify the statistical analysis.

Author response:

We apologize for this omission. The data were from three independent cultures. Individual data points, mean (bars) and standard deviation (error bars) are shown. We have revised the figure legend to include this crucial information (lines 939-940).

9. Line 323 – the authors state that studies are needed to determine whether JFBs are permissive to SARS-CoV-2 infection in vivo, yet newly added text (lines 285 – 287) states that a recent preprint describing in vivo infection experiments in JFBs indicates that it leads to an abortive infection of the intestine without development of clinical disease (ref 66). Please resolve this apparent contradiction.

Author response:

Thanks for pointing out this discrepancy. We have changed this statement (now line 329-330) as follows: “Further experiments are needed to evaluate at which stage of the viral replication cycle SARS-CoV-2 replication stalls in the JFB organoid model and upon in vivo infection of JFBs.”

10. “data is not shown” appears in two places (line 137 and line 186).

Author response:

To address the reviewer’s concern about data not shown, we have removed the respective statements from the manuscript, since they did not provide any essential information.